# STRETCH TRANSFORMATION FOR TABULAR DATA

## ABSTRACT

Tabular data presents unique challenges for deep learning due to its heterogeneous nature, where features exhibit diverse distributions, scales, and statistical properties. Although recent advances have achieved strong performance on tabular benchmarks, *feature transformation*, a critical preprocessing step, remains largely unsupervised despite the availability of target information during training. We introduce the *stretch transformation framework*, which formulates feature preprocessing as an optimization problem to make the target function smoother and thus more learnable. Our framework has two variants: (1) *unsupervised stretch*, which uniformly redistributes feature density via minimax optimization, and (2) *supervised stretch*, which is the first method to systematically leverage target information for numeric features by minimizing the target function's Dirichlet energy in the transformed space. Our theoretical analysis reveals fundamental connections to existing methods, as unsupervised stretch explains why empirical CDF transformation can improve learning despite being label-agnostic, and supervised stretch generalizes target encoding with principled regularization for numeric features. Comprehensive experiments on 38 datasets from the TALENT benchmark demonstrate that supervised stretch consistently outperforms all baselines. These results show that explicitly optimizing for target function smoothness is a powerful and underexplored strategy for tabular deep learning.

## 1 INTRODUCTION

Tabular data presents unique challenges for machine learning due to its inherent heterogeneity. Unlike images with uniform pixel grids or text with discrete vocabularies, tabular features combine measurements of fundamentally different natures—binary flags, counts, percentages, monetary values, and heavy-tailed distributions—each with distinct scales and statistical properties (Grinsztajn et al., 2022). This heterogeneity creates a challenging optimization landscape for neural networks, which typically assume well-conditioned inputs (Ioffe & Szegedy, 2015).

While tree-based methods naturally handle diverse distributions through split-based decisions, neural networks require careful preprocessing to achieve competitive performance (Ye et al., 2024; Liu et al., 2024). Thus, *feature transformation* step is needed to bridge the gap between raw heterogeneous data and model-friendly features. Commonly used transformations include standardization, power transformations (Yeo & Johnson, 2000), quantile transformations (Pedregosa et al., 2011), and Piecewise Linear Encoding (PLE) (Gorishniy et al., 2022). Remarkably, although target information (e.g., labels) is typically available during training, virtually all existing methods are unsupervised. PLE is a notable exception, but it only uses targets to set bin boundaries, not to optimize the transformation within those bins to improve learnability.

We introduce the *stretch transformation framework*, which systematically incorporates target information to create more learnable representations. Our key insight is that feature transformation should not merely normalize distributions but actively make the target function smoother in the transformed space. We present two complementary approaches:

**Supervised stretch** is, to our knowledge, the *first method to use target information for creating smoother target functions* through transformation. By minimizing the Dirichlet energy of the target function in the transformed space, we derive optimal width allocations that concentrate more resolution in regions where the target varies rapidly. This principled approach achieves remarkable empirical gains, demonstrating the untapped potential of supervised feature transformation.

**Unsupervised stretch** provides a robust fallback when target information is unavailable or unreliable. Using minimax optimization, it maximizes the worst-case sample separation, effectively creating a piecewise linear approximation of the empirical CDF transformation. This variant matches or exceeds PLE's performance while requiring only $O(1)$ memory per feature instead of $O(T)$.

Beyond practical improvements, our framework offers theoretical explanations for existing empirical observations. We show that unsupervised stretch explains why CDF transformation can reduce frequency content and improve learnability (Beyazit et al., 2023), despite being label-agnostic. Similarly, supervised stretch reveals deep connections to target encoding, providing a theoretical justification for why target-based transformations can be effective while offering principled regularization through the number of bins $T$. **In summary, our contributions are:**

- We introduce supervised stretch, the first systematic framework that uses target information to optimize numeric feature transformations by maximizing target smoothness.
- We propose unsupervised stretch, a memory-efficient alternative to PLE that achieves comparable or superior performance without dimensional expansion.
- Our theoretical analysis connects supervised stretch to target encoding and unsupervised stretch to CDF transformation, providing principled explanations for their empirical effectiveness.
- Comprehensive experiments on 38 datasets show that supervised stretch consistently outperforms all baselines, with particularly strong gains in regression tasks.

These results challenge the prevailing paradigm of unsupervised feature transformation in tabular deep learning and open new avenues for supervised preprocessing methods.

## 2 RELATED WORK

**Tabular Machine Learning** is a rapidly evolving field (van Breugel & van der Schaar, 2024; Borisov et al., 2024; Shwartz-Ziv & Armon, 2021; Hancock & Khoshgoftaar, 2020). Recent work has proposed various neural architectures for tabular data, including attention-based models (Gorishniy et al., 2021; Arik & Pfister, 2021) and MLP-based models (Gorishniy et al., 2022; Holzmüller et al., 2024; Gorishniy et al., 2025), achieving strong performance on standard benchmarks (Erickson et al., 2025). Beyond architectural innovations, several studies have explored pretraining for large-scale tabular datasets (Kim et al., 2024), while recent advances in in-context learning have enabled foundation models to perform well on small datasets without task-specific finetuning (Hollmann et al., 2023; 2025; Qu et al., 2025).

**Feature Transformation for Tabular Data.** Effective preprocessing is crucial for tabular machine learning, as heterogeneous features span binary indicators, counts, and heavy-tailed continuous variables with disparate scales (Grinsztajn et al., 2022). This complexity necessitates feature transformation strategies tailored to tabular data. While tree-based models (e.g., Random Forests, Gradient Boosted Trees) are invariant to monotonic transformations (Chen & Guestrin, 2016), neural networks often require standardized inputs to stabilize optimization (Ioffe & Szegedy, 2015).

Standardization (z-score normalization) remains the most common preprocessing technique, mapping features to zero mean and unit variance (Pedregosa et al., 2011). However, it only rescales and shifts distributions without addressing skewness or outliers. Power transformations address non-normality: the Yeo-Johnson transformation (Yeo & Johnson, 2000) extends Box-Cox (Box & Cox, 1964) to handle negative values, producing more Gaussian-like features, which benefits neural networks and has been widely adopted in tabular machine learning (Hollmann et al., 2023). Piecewise linear encoding (PLE) (Gorishniy et al., 2022) discretizes the distribution into quantile-based bins and encodes each into a vector, which has shown effectiveness in tabular tasks.

**Neural Network Inductive Biases.** Despite being universal approximators (Cybenko, 1989; Hornik, 1991), neural networks exhibit specific inductive biases that can influence their performance in tabular data. For instance, the spectral bias phenomenon, whereby ReLU networks preferentially learn low-frequency functions before high-frequency components, has been well-documented (Rahaman et al., 2019; Xu et al., 2019). Interestingly, Beyazit et al. (2023) observe that tabular data contains substantially higher frequency components compared to natural images. Unlike images defined on a fixed pixel grid, numeric features imply continuity, allowing samples to be arbitrarily close to each other. This characteristic creates unbounded high-frequency components that are noto-

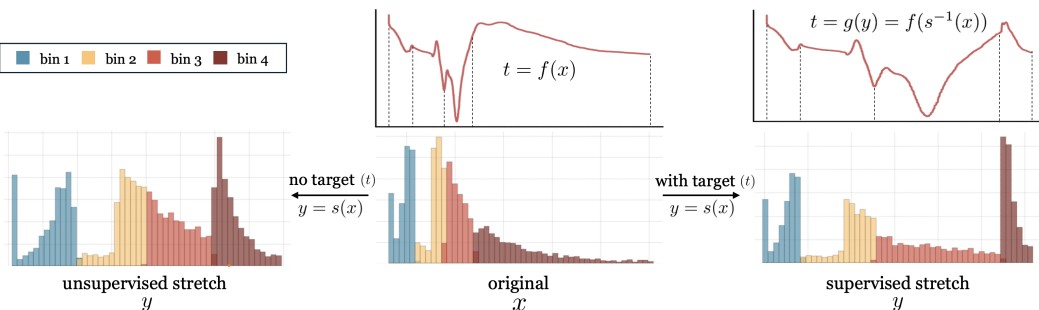

Figure 1: Overview: Stretch Transformation Framework. **(Center)** An original feature $x$ with a complex target function $f(x)$, partitioned into four quantile bins (colors). **(Left)** Unsupervised stretch uniformly redistributes feature density. **(Right)** Supervised stretch uses target information to allocate more *stretch* to bins where the target function varies rapidly (e.g., bin 3). This creates a new target function $g(y)$ in the transformed space that is significantly smoother and thus easier for a neural network to learn.

riously difficult for neural networks to learn due to spectral bias. This observation provides one lens through which to understand the empirical performance gap between neural networks and tree-based methods on tabular benchmarks.

## 3 STRETCH FEATURE TRANSFORMATION

Denote a tabular dataset by $\mathcal{D} = \{(\mathbf{x}_i, t_i)\}_{i=1}^n$, where $\mathbf{x}_i \in \mathbb{R}^d$ are features and $t_i$ is the corresponding target.[1] While targets are available during training, feature transformations may leverage this information (supervised) or ignore it (unsupervised), though most existing methods are unsupervised. A feature transformation $s : \mathcal{X} \to \mathcal{Y}$ preprocesses the raw features, creating the learning pipeline:

$$\mathbf{x} \xrightarrow{s} \mathbf{y} = s(\mathbf{x}) \xrightarrow{g_\theta} \hat{t} = g_\theta(\mathbf{y}) \tag{1}$$

For an invertible transformation with the inverse $h = s^{-1}$, the target function in the transformed space becomes $g(y) = \mathbb{E}[t|y] = f(h(y))$ where $f(x) = \mathbb{E}[t|x]$ is the original target function. The neural network $g_\theta$ learns to approximate $g$ in the transformed space. The goal is to find $s$ such that $g = f \circ h$ is easier for neural networks to learn than the original $f$.

Feature transformations can be characterized along several axes: they may be unsupervised (using only feature statistics, e.g., standardization, quantile transformation) or supervised (leveraging target information); dimension-preserving ($d' = d$) or dimension-expanding ($d' > d$, as in PLE); and marginal, operating on features independently, or joint, modeling feature interactions. These properties can be combined as well, for instance, distribution-shaping followed by standardization.

For numeric features, desirable properties of a transformation $s$ include: (1) *monotonicity*, preserving the natural ordering to avoid arbitrary permutations that could destroy meaningful relationships; (2) *bounded output range*, e.g., $[0, 1]$ or $[-1, 1]$, to ensure inputs are well-scaled for neural networks.

Our stretch transformation framework, introduced in the next section, provides both unsupervised and supervised variants that satisfy these requirements through a piecewise linear design. The framework operates marginally on each feature while preserving dimensionality. For the supervised variant, target information is used *only* during transformation design; the learned transformation is then fixed and applied identically to all data.

### 3.1 STRETCH TRANSFORMATION SETUP

We introduce a feature transformation that rescales different regions of the input via a monotone piecewise linear map. For a scalar feature $x \in \mathbb{R}$ with range $[x_{\min}, x_{\max}]$, we first partition the

---

[1]For simplicity, we write $t_i$ as the target. Our framework is directly applicable to both scalar targets (e.g., class labels, single value regression) and multidimensional vector targets ($\mathbf{t}_i \in \mathbb{R}^k$).

domain into $T$ intervals with boundaries $\mathbf{b} = \{b_0, b_1, \ldots, b_T\}$ where $b_0 = x_{\min}$ and $b_T = x_{\max}$. We adopt quantile-based binning as the default partition, ensuring approximately equal sample counts per bin. This process groups the samples $\{x_i\}_{i=1}^n$ (assumed to be sorted) into $T$ consecutive bins. We denote the set of sample indices $i$ whose corresponding feature value $x_i$ falls into the $t$-th bin, $[b_{t-1}, b_t)$, as $\mathcal{I}_t$. The transformation $s : \mathbb{R} \to [0, 1]$ is then defined by

$$s(x) = c_{t-1} + \frac{x - b_{t-1}}{b_t - b_{t-1}} \cdot w_t, \qquad x \in [b_{t-1}, b_t) \,, \tag{2}$$

where $w_t > 0$ is the width allocated to interval $t$, satisfying $\sum_{t=1}^T w_t = 1$. The term $c_t := \sum_{i=1}^t w_i$ (with $c_0 = 0$) denotes the cumulative width. This form is strictly increasing (hence order-preserving and bijective onto $[0, 1]$). The slope within each interval is

$$s'(x) = \frac{w_t}{b_t - b_{t-1}} \,, \quad \forall x \in [b_{t-1}, b_t), \ t \in [T] \,. \tag{3}$$

Thus, by choosing $\{w_t\}$, we can control the local stretch (large $w_t$) or squeeze (small $w_t$) of each region. **The central design problem is how to set the width vector $\{w_t\}_{t=1}^T$, which we derive from a principled optimization objective.**

### 3.2 Optimization Objective: Kernel Smoothness Maximization

The primary goal of feature transformation is to make the target function easier to learn in the transformed space. Let $f(x) := \mathbb{E}[t|x]$ denote the marginal target function (vector-valued for classification), and $g(y) = f(h(y))$ be the target function in the transformed space where $h = s^{-1}$. A smoother $g$ in the $y$-space is easier for neural networks to learn due to their spectral bias toward low-frequency functions (Xu et al., 2019; Rahaman et al., 2019). Thus, we seek transformations that maximize the smoothness of $g$. Following the principle of kernel smoothness maximization, we formalize this objective by minimizing the Dirichlet energy of the target function. We derive this connection below, with the full derivation provided in Appendix A.1. To quantify smoothness, we consider the kernel correlation with a Gaussian RBF kernel $k_\sigma(y, z) = \exp(-(y - z)^2/(2\sigma^2))$:

$$\langle g, T_\sigma g \rangle = \int_0^1 \int_0^1 g(y) k_\sigma(y, z) g(z) \, dy \, dz \,, \tag{4}$$

where $T_\sigma$ is the integral operator associated with $k_\sigma$. In the small bandwidth regime ($\sigma \to 0$), the heat kernel expansion (Varadhan, 1967; Molchanov, 1975) gives $T_\sigma = e^{\frac{\sigma^2}{2}\partial_{yy}} = I + \frac{\sigma^2}{2}\partial_{yy} + O(\sigma^4)$. To isolate the term independent of smoothness, we use the centered version:

$$\langle g, (T_\sigma - I)g \rangle = \langle g, \frac{\sigma^2}{2}\partial_{yy}g \rangle + O(\sigma^4) = -\frac{\sigma^2}{2} \int_0^1 \|g'(y)\|_2^2 \, dy + O(\sigma^4), \tag{5}$$

where the last equality follows from integration by parts. Therefore, maximizing kernel smoothness is equivalent to minimizing the Dirichlet energy:

$$\min_h \ \mathcal{E}[g] := \int_0^1 \|g'(y)\|_2^2 \, dy \,. \tag{6}$$

Recall that our stretch transformation allocates width $w_t$ to bin $t$, controlling how much of the transformed space $[0, 1]$ is dedicated to each region. Given samples $\{x_i\}_{i=1}^n$ with transformed positions $y_i = s(x_i)$, approximating the continuous function $g$ by its piecewise linear interpolant yields the discrete Dirichlet energy:

$$\mathcal{E}_{\mathrm{disc}} = \sum_{i=1}^{n-1} \frac{\|\Delta f_i\|_2^2}{\Delta y_i}, \quad \text{where} \quad \Delta f_i := f(x_{i+1}) - f(x_i), \quad \Delta y_i := y_{i+1} - y_i. \tag{7}$$

This objective depends on both the target increments $\Delta f_i$ and the space in the transformed space $\Delta y_i$. The spacings $\Delta y_i$ are determined by our width allocations: within bin $t$ with width $w_t$, the spacings sum to $w_t$. The key question then is how to choose $\{w_t\}$ to minimize this energy, which we answer in two distinct settings of unsupervised and supervised.

### 3.3 UNSUPERVISED STRETCH: UNIFORM ALLOCATION

Without the target information, we cannot directly evaluate $\Delta f_i$ in Eqn. (7). Instead, we adopt a minimax principle and minimize the worst-case Dirichlet energy over all target functions with bounded total variation. Specifically, assuming bounded local variation $\|\Delta f_i\|_2^2 \leq C$ for all $i$, the worst-case Dirichlet energy for spacing allocation $\{\Delta y_i\}$ becomes

$$\max_{\{\Delta f_i\}} \sum_{i=1}^{n-1} \frac{\|\Delta f_i\|_2^2}{\Delta y_i} \quad \text{s.t.} \quad \|\Delta f_i\|_2^2 \leq C, \ \forall i. \tag{8}$$

The maximizer is straightforwardly $\|\Delta f_i\|_2^2 = C$ for all $i$, leading to the optimization problem (see Appendix A.2 for detailed derivation):

$$\min_{\{\Delta y_i\}} \sum_{i=1}^{n-1} \frac{1}{\Delta y_i} \quad \text{s.t.} \quad \sum_{i=1}^{n-1} \Delta y_i = 1 \ . \tag{9}$$

Within our binning framework with $T$ quantile bins containing $n_t \approx n/T$ samples each, the optimal allocation is then uniform both within and across bins, yielding:

$$w_t^\star = \frac{1}{T} \ , \qquad \forall t \in \{1, \ldots, T\} \ . \tag{10}$$

**Connection to empirical CDF.** With this uniform allocation, unsupervised stretch becomes a piecewise linear approximation of the empirical CDF transformation. As $T \to n$, our method converges to the exact empirical CDF, which maps each sample to its normalized rank. This insight provides a theoretical justification for the empirical finding in Beyazit et al. (2023) that CDF transformation reduces frequency content and improves learnability. By maximizing the minimum spacing between samples, we effectively smooth the target function, reducing the high-frequency components. However, as noted in Beyazit et al. (2023), reducing high-frequency components does not guarantee a better performance, since excessive smoothing can eliminate important signals. The parameter $T$ provides a natural trade-off: a small $T$ preserves more of the original distribution's structure, while a large $T$ approaches full CDF transformation. This controllable interpolation between preserving distributional features and achieving uniform density is a key advantage of our framework.

**Connection to Piecewise Linear Encoding (PLE).** Although unsupervised stretch and PLE (Gorishniy et al., 2022) appear fundamentally different—scalar versus $T$-dimensional outputs—they share an underlying geometric structure. PLE maps a sample $x$ in bin $t$ to:

$$\text{PLE}(x) = [1, \ldots, 1, \frac{x - b_{t-1}}{b_t - b_{t-1}}, 0, \ldots, 0] \in \mathbb{R}^T \ . \tag{11}$$

This creates a piecewise linear path in $\mathbb{R}^T$ from origin to $(1, \ldots, 1)$. The arc length along this path is

$$L(x) = (t-1) + \frac{x - b_{t-1}}{b_t - b_{t-1}} = T \cdot \text{Unsupervised-Stretch}(x) \ . \tag{12}$$

Thus, unsupervised stretch precisely parameterizes the PLE manifold by normalized arc length. Both achieve identical quantile-based density redistribution, differing only by coordinate representation. This equivalence explains their similar empirical performance (Section 4.2), while unsupervised stretch offers significant computational advantages: $O(1)$ versus $O(T)$ memory per feature and no dimensional expansion.

### 3.4 SUPERVISED STRETCH: TARGET-INFORMED ALLOCATION

When target information is available, we can optimize the discrete Dirichlet energy in Eqn. (7) directly. However, substituting the piecewise linear map proves to be numerically unstable, as the resulting objective is highly sensitive to the original feature spacings $\Delta x_i$ (see Appendix A.3 for a detailed discussion). To overcome this, we instead optimize a robust lower bound on the energy.

We partition the samples into $T$ bins. For bin $t$ with width $w_t = \sum_{i \in \mathcal{I}_t} \Delta y_i$, let $S_t := \sum_{i \in \mathcal{I}_t} \|\Delta f_i\|_2$ be the bin-wise total variation of the target. For fixed $w_t > 0$, minimizing the within-bin contribution of Eqn. (7) over $\{\Delta y_i\}_{i \in \mathcal{I}_t}$ subject to $\sum_{i \in \mathcal{I}_t} \Delta y_i = w_t$ yields a lower bound:

$$\sum_{i \in \mathcal{I}_t} \frac{\|\Delta f_i\|_2^2}{\Delta y_i} \geq \frac{S_t^2}{w_t} \ , \tag{13}$$

with equality at $\Delta y_i \propto \|\Delta f_i\|_2$ within each bin. Then, the global optimization over $\{w_t\}$ becomes

$$\min_{\{w_t > 0\}} \sum_{t=1}^{T} \frac{S_t^2}{w_t} \quad \text{s.t.} \quad \sum_{t=1}^{T} w_t = 1 \ . \tag{14}$$

This convex problem has the closed-form solution:

$$w_t^{\star} = \frac{S_t}{\sum_{u=1}^{T} S_u} . \tag{15}$$

Intuitively, in the transformed space, we allocate a larger space to the regions where the target function varies rapidly, effectively equalizing the slope magnitudes across bins and creating a smoother target function in the transformed space.

**Connection to target encoding.** Supervised stretch connects naturally to target encoding when $T = n$ (one bin per unique value). In this limit, the optimal width for the interval $[x_i, x_{i+1}]$ becomes:

$$w_i = \frac{|f_{i+1} - f_i|}{\sum_{k=1}^{n-1} |f_{k+1} - f_k|} \ . \tag{16}$$

This is remarkably similar to applying min-max scaling to target encoding. Standard target encoding maps $x_i \mapsto f_i = \mathbb{E}[t|x = x_i]$, and if we scale these values to $[0, 1]$, we get essentially the same transformation for monotonic targets. The key insight is that both methods fundamentally use target variation to guide the transformation—target encoding does it directly, while supervised stretch achieves it via Dirichlet energy minimization. Our framework thus provides a theoretical justification for why target-based transformations are effective: they implicitly smooth the target function in the transformed space.

While target encoding is widely applied to categorical features, its use for numeric features has been limited, partly due to concerns about overfitting or interpretability. The proposed supervised stretch offers a regularized, theoretically grounded approach to incorporating target information for numeric features, with the number of bins $T$ serving as a natural regularization parameter.

**Out-of-fold estimation.** In practice, to prevent information leakage, we use $K$-fold cross-validation with adaptive Nadaraya-Watson kernel regression to obtain $\widehat{f}(x_i)$ for each sample without using its own target value (Nadaraya, 1964; Watson, 1964) (details in Appendix C.1). The bin widths are computed using $\Delta \widehat{f}_i$ in place of $\Delta f_i$.

## 4 EXPERIMENTS

We evaluate the proposed stretch transformations against established baselines across comprehensive tabular benchmarks, assessing their impact on diverse neural architectures. Our experiments demonstrate consistent advantages of supervised stretch, particularly in regression tasks, while unsupervised stretch provides a memory-efficient alternative to PLE. Detailed experimental configurations and additional results are provided in Appendix C.3.

### 4.1 EXPERIMENTAL SETUP

**Datasets and Models.** We use the TALENT benchmark suite (Ye et al., 2024; Liu et al., 2024), focusing on the *Tiny Benchmark 1* collection of 26 classification and 12 regression datasets. This benchmark covers diverse tabular learning tasks with varying dataset sizes, feature types, and target distributions. Categorical features are encoded as indices by default, with RealMLP internally converting them to one-hot encoding as per its design (Holzmüller et al., 2024). Our investigation focuses on numeric feature transformations. Detailed characteristics of these datasets, including the breakdown of numeric versus categorical feature proportions, sample sizes, and feature counts, are provided in Appendix D.

We evaluate five representative neural architectures for tabular data: FT-Transformer (FTT) (Gorishniy et al., 2021), a standard MLP, MLP-PLR (Gorishniy et al., 2022), RealMLP (Holzmüller et al., 2024), and a ResNet (He et al., 2015). Experiments span 38 datasets, yielding 190 dataset-model combinations. To ensure a fair comparison, we disable RealMLP's built-in

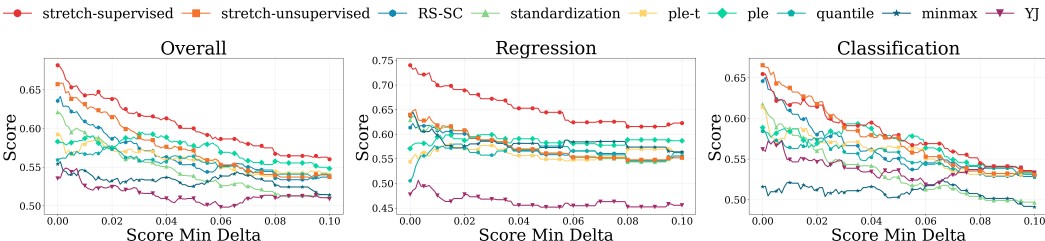

Figure 2: Sensitivity analysis of significance thresholds. Average normalized score is plotted against a fixed minimum separation $\delta$. Larger $\delta$ values treat small gaps as ties (score 0.5), leading all methods to converge toward 0.5. Unlike the adaptive threshold used in Table 1, this analysis varies $\delta$ across fixed values. Critically, the relative performance ranking remains stable across all reasonable thresholds well before convergence, confirming robustness to threshold choice. Key takeaways: supervised stretch clearly dominates in regression, and both supervised and unsupervised stretch show top-tier performance in classification. **Abbreviations**: PLE-T (PLE with Tree-based binning), Quantile (Quantile Gaussian), RS-SC (RobustScale+SmoothClip), and YJ (Yeo Johnson).

`RobustScale+SmoothClip` (RS-SC) preprocessing when applying other transformations. The original RealMLP configuration, with RS-SC enabled, is included separately to benchmark the RS-SC method itself.

**Transformation Methods.** We compare our proposed supervised and unsupervised stretch against seven established baselines: **standardization** (z-score normalization), **Yeo-Johnson (YJ)** power transformation (Yeo & Johnson, 2000), **quantile transformation** to a Gaussian distribution, **min-max** scaling to $[0, 1]$, **RobustScale+SmoothClip (RS-SC)** as originally used in RealMLP (Holzmüller et al., 2024), **Piecewise Linear Encoding (PLE)** (Gorishniy et al., 2022), and **PLE with Tree-based binning (PLE-T)**. PLE is an unsupervised method that typically uses quantile-based binning, whereas PLE-T is a supervised variant that leverages a tree-based model to create bins informed by the target variable. To our knowledge, PLE-T is the only existing supervised transformation for numeric features in the literature and thus serves as our primary supervised baseline. For transformations that only adjust distribution shape (Yeo-Johnson, quantile, PLE, PLE-T, and stretch variants), we apply standardization afterward to ensure consistent scaling. Implementation details regarding computational constraints are provided in Appendix C.2.

**Evaluation Protocol.** Following the benchmark protocols (Ye et al., 2024; Liu et al., 2024), we conduct Bayesian optimization using Optuna (Akiba et al., 2019) with 100 trials for each dataset-model-transformation combination, jointly tuning model and transformation hyperparameters (e.g., number of bins for stretch and PLE). Each optimal configuration is evaluated with 15 random seeds. We use accuracy for classification and $R^2$ (clipped to $[0, 1]$) for regression as primary metrics.

**Performance Normalization.** Following Grinsztajn et al. (2022), we apply min-max normalization per dataset-model pair to enable fair comparison across heterogeneous datasets:

$$\text{score}_{d,m,t} = \frac{\text{metric}_{d,m,t} - \min_t \text{metric}_{d,m,t}}{\max_t \text{metric}_{d,m,t} - \min_t \text{metric}_{d,m,t}}, \tag{17}$$

where $d, m, t$ index datasets, models, and transformations, respectively. We also introduce a *significance filter* to focus on chases where the choice of transformation has a meaningful impact. For each dataset-model pair $(d, m)$, we set a threshold $\delta_{d,m}$ equal to the *median* of standard errors, computed for each transformation across 15 seeds. If the gap between the best and worst transformation ($\max_t \text{metric}_{d,m,t} - \min_t \text{metric}_{d,m,t}$) is smaller than this threshold $\delta_{m,t}$, the choice of transformation is deemed inconsequential, and we assign a normalized score of 0.5 to all transformations, denoting a tie. These inconsequential pairs are excluded when calculating the overall Avg. Accuracy and Avg. $R^2$ in Table 1, ensuring the reported averages reflect tasks where preprocessing matters. Finally, note that this adaptive threshold is used in Table 1, whereas the sensitivity analysis in Figure 2, by design, explores a range of fixed $\delta$ values.

**Pairwise Comparisons.** For a robust head-to-head comparison (Figure 3), we compute pairwise win-rates using a strict, statistically-aware criterion. For a dataset-model combination, a transformation $t_1$ is considered to win against $t_2$ only if its performance exceeds that of $t_2$ by more than

Table 1: Performance of transformations grouped by model. Normalized Score is computed with adaptive significance filter: for each dataset-model pair, if the gap between the best and worst method is smaller than the *median standard error* across all methods, the pair is considered a tie (score 0.5). Tie pairs are excluded from the 'Avg. Accuracy' and 'Avg. $R^2$' calculations. For detailed metrics including standard deviations, see Appendix C.3. For performance stratified by dataset feature composition, see Appendix E. **Abbreviations**: Sup. (Stretch Supervised), Unsup. (Stretch Unsupervised).

| Metric | Model | Transformations | | | | | | | | |
|--------|-------|------|--------|--------|------|-------|----------|-------|----------|------|
| | | Sup. | Unsup. | Minmax | PLE | PLE-T | Quantile | RS-SC | Standard | YJ |
| Overall Score | ftt | **0.6942** | 0.6543 | 0.3824 | 0.6405 | 0.6461 | 0.6559 | 0.5643 | 0.6214 | 0.5536 |
| | mlp | 0.6343 | 0.6275 | 0.5600 | **0.6835** | 0.6182 | 0.5526 | 0.6653 | 0.6423 | 0.5790 |
| | mlp_plr | 0.6238 | **0.7347** | 0.6037 | 0.4513 | 0.5763 | 0.5673 | 0.6091 | 0.6443 | 0.5802 |
| | realmlp | **0.7078** | 0.6348 | 0.6321 | 0.5457 | 0.5967 | 0.5566 | 0.6602 | 0.5919 | 0.4482 |
| | resnet | **0.7237** | 0.6610 | 0.6175 | 0.5999 | 0.5294 | 0.4854 | 0.6989 | 0.6205 | 0.5160 |
| Cls. Score | ftt | 0.6815 | 0.6391 | 0.3434 | 0.6711 | **0.6929** | 0.6838 | 0.5419 | 0.6003 | 0.5598 |
| | mlp | 0.5738 | 0.6322 | 0.4900 | **0.6806** | 0.6324 | 0.5509 | 0.6480 | 0.6525 | 0.5679 |
| | mlp_plr | 0.6134 | **0.7265** | 0.5379 | 0.4578 | 0.5675 | 0.6077 | 0.6473 | 0.6354 | 0.6215 |
| | realmlp | **0.7074** | 0.6784 | 0.6799 | 0.5051 | 0.6110 | 0.5871 | 0.6961 | 0.6112 | 0.4656 |
| | resnet | 0.6809 | 0.6702 | 0.5473 | 0.6180 | 0.5570 | 0.5033 | **0.7054** | 0.5914 | 0.5763 |
| Reg. Score | ftt | **0.7217** | 0.6872 | 0.4669 | 0.5743 | 0.5446 | 0.5953 | 0.6129 | 0.6671 | 0.5400 |
| | mlp | **0.7653** | 0.6173 | 0.7117 | 0.6897 | 0.5874 | 0.5561 | 0.7028 | 0.6203 | 0.6030 |
| | mlp_plr | 0.6464 | **0.7525** | 0.7463 | 0.4372 | 0.5954 | 0.4796 | 0.5262 | 0.6635 | 0.4907 |
| | realmlp | **0.7089** | 0.5403 | 0.5287 | 0.6337 | 0.5656 | 0.4907 | 0.5824 | 0.5501 | 0.4104 |
| | resnet | **0.8165** | 0.6409 | 0.7697 | 0.5606 | 0.4697 | 0.4464 | 0.6847 | 0.6838 | 0.3855 |
| Avg Acc | ftt | **0.8274** | 0.8244 | 0.8052 | 0.8262 | 0.8261 | 0.8272 | 0.8191 | 0.8162 | 0.8212 |
| | mlp | 0.8209 | 0.8245 | 0.8152 | **0.8275** | 0.8217 | 0.8160 | 0.8235 | 0.8193 | 0.8222 |
| | mlp_plr | 0.8257 | 0.8264 | 0.8210 | 0.8212 | 0.8213 | **0.8271** | 0.8228 | 0.8212 | 0.8214 |
| | realmlp | 0.8373 | 0.8376 | **0.8387** | 0.8331 | 0.8345 | 0.8370 | 0.8383 | 0.8327 | 0.8339 |
| | resnet | **0.8257** | 0.8239 | 0.8145 | 0.8236 | 0.8232 | 0.8163 | 0.8246 | 0.8189 | 0.8201 |
| Avg $R^2$ | ftt | **0.6911** | 0.6809 | 0.6663 | 0.6799 | 0.6671 | 0.6813 | 0.6726 | 0.6815 | 0.6669 |
| | mlp | **0.6304** | 0.6060 | 0.6279 | 0.6077 | 0.5877 | 0.6190 | 0.6192 | 0.5909 | 0.5453 |
| | mlp_plr | 0.6983 | 0.6963 | 0.6591 | 0.6859 | **0.7020** | 0.6689 | 0.6809 | 0.6754 | 0.6489 |
| | realmlp | **0.7362** | 0.6846 | 0.6791 | 0.7357 | 0.7234 | 0.7066 | 0.6888 | 0.6858 | 0.6832 |
| | resnet | 0.6507 | 0.6312 | **0.6547** | 0.6386 | 0.6307 | 0.6254 | 0.6399 | 0.6409 | 0.5421 |

their standard errors: $\text{metric}_{t_1} > \text{metric}_{t_2} + \max(\sigma_{t_1}, \sigma_{t_2})$. This criterion ensures that a declared win represents a statistically meaningful performance gap, rather than an artifact of noise. The final win-rate is the percentage of wins across all 190 combinations, where ties (if neither method satisfies the win condition) are scored as 0.5.

## 4.2 RESULTS AND ANALYSIS

Our analysis draws on three complementary perspectives. First, Figure 2 presents a sensitivity analysis using a range of fixed significance thresholds ($\delta$) to demonstrate the robustness of our findings. Second, Table 1 provides a detailed breakdown for each model using an adaptive threshold based on the median standard error of each specific evaluation. This reveals how different architectures interact with each transformation under a statistically grounded criterion. Finally, the pairwise win rate heatmap in Figure 3 offers direct head-to-head comparisons to identify the most consistently superior methods. Together, these analyses lead to three key findings:

**1. Supervised stretch achieves dominant and robust performance.** Across all evaluations, supervised stretch stands out as the strongest method. Its advantage is most striking in regression, where it maintains a substantial margin over all competitors regardless of the significance threshold $\delta$ (Figure 2, middle panel). While the competition is tighter in classification, it remains in the top tier. The model-specific results in Table 1 confirm this, as supervised stretch achieves the highest Overall Score for three of five models and the highest Regression Score for four of five models. The win-rate analysis (Figure 3) further shows that it statistically outperforms every alternative. This dominance is further quantified in our detailed pairwise comparisons (Appendix B), particularly in regression, where it secures decisive win-loss records of **18-7** against PLE, **28-9** against PLE-T, and **14-4** against standardization. These results strongly validate our hypothesis that smoothing target functions via kernel smoothness maximization significantly improves neural network performance.

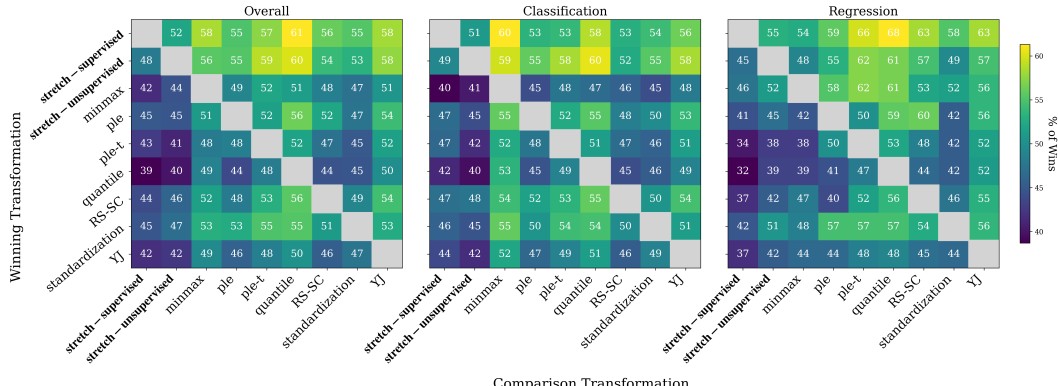

Figure 3: Pairwise win-rates between transformations, grouped by task. Each cell $(i, j)$ shows the percentage of times transformation $i$ (row) statistically outperforms transformation $j$ (column). Supervised stretch dominates across all methods and tasks. Unsupervised stretch is the strongest unsupervised baseline, second only to its supervised counterpart, with particularly competitive performance in classification. For detailed win-loss-tie analysis against the strongest unsupervised (PLE, Standardization) and supervised (PLE-T) baselines, see Appendix B.

**2. Unsupervised stretch is the strongest target-agnostic method.** Unsupervised stretch consistently ranks as the clear runner-up, establishing itself as the best transformation that does not require target information. It outperforms all other methods except its supervised counterpart (Figure 3) and is especially competitive in classification (Figure 2, right). This strength in classification is starkly evident in direct comparisons (Appendix B), with impressive records of **38–25** vs. PLE, **38–18** vs. PLE-T, and **26–13** vs. Standardization. It even surpasses the supervised variant against some baselines. Model-specific results also support its strength; for instance, it is the best-performing transformation for MLP-PLR (Table 1). This practical advantage is compounded by its efficiency: its per-feature memory overhead is merely $O(1)$ compared to PLE's $O(T)$, and it avoids computational bottlenecks that can make PLE impractical for large datasets.

**3. Transformation preferences vary by architecture and task type.** Although supervised stretch demonstrates clear overall superiority, our model-specific results reveal that architectural preferences exist. For example, while supervised stretch is the best generalist, unsupervised stretch is the top choice for the MLP-PLR architecture. More importantly, task type drives the largest differences, as regression consistently benefits more from advanced transformations than classification. This is most apparent with supervised stretch, which yields dramatic gains in continuous prediction tasks (Figure 2, middle). These findings suggest that explicitly engineering smoother mappings from features to the target is an especially valuable strategy for regression, an underexplored direction in a literature largely focused on classification.

**4. Non-uniform marginal signals necessitate smoothing, especially for regression.** To understand the underlying mechanism driving these performance gains, we conducted a systematic analysis of marginal signal structures in Appendix C.3. We found that real-world tabular features are characterized by non-uniform marginal distributions with localized regions of sharp variation (high relative marginal slopes), rather than being flat or uniform. This structural heterogeneity creates high-frequency components that are difficult for neural networks to learn due to spectral bias. Supervised stretch explicitly identifies and smooths these high-gradient regions. This mechanism is particularly critical for regression, where the target space is continuous and unbounded, explaining why the benefits of our method are more pronounced in regression tasks compared to classification, where the goal is merely to find a decision boundary.

## 5 CONCLUSION

This paper introduced the **stretch transformation framework**, a principled approach to feature preprocessing for tabular deep learning. By framing transformation as an optimization problem that minimizes the Dirichlet energy of the target function, we developed two powerful methods. **Supervised stretch**, the first method to systematically exploit target information for creating smoother target functions, achieves state-of-the-art performance. **Unsupervised stretch** maximizes worst-

case separation through uniform density redistribution. Our analysis also explains the success of existing techniques. Unsupervised stretch clarifies why empirical CDF transformation improves learning despite being target-agnostic, and reveals its equivalence to PLE through arc length parameterization. Further, we show that supervised stretch and target encoding both leverage target variation, but our framework provides principled regularization through the number of bins. Empirically, supervised stretch achieved remarkable gains across 38 datasets, validating our hypothesis that target-aware transformation significantly enhances neural network performance. These results challenge the prevailing paradigm of unsupervised preprocessing in tabular deep learning.

**Limitations and Future Work.** Our framework's design could be enhanced by exploring smoother, spline-based alternatives, and adaptive binning strategies. Theoretically, the analysis could be generalized beyond the small-$\sigma$ RBF kernel approximation, and practically, the framework could be extended from marginal transformations to model feature interactions.

In summary, this work establishes that supervised feature transformation is a powerful yet underexplored tool for tabular deep learning, opening new avenues for principled preprocessing methods that fully exploit all available information.

## ETHICS STATEMENT

The authors have read and adhere to the ICLR Code of Ethics. Our work is foundational in nature, focusing on a new algorithmic approach for feature preprocessing in tabular data. The research does not involve human subjects, sensitive data, or personally identifiable information. All experiments were conducted on publicly available benchmark datasets from the TALENT suite. We do not foresee any direct negative societal impacts or ethical concerns arising from the proposed methodology.

## REPRODUCIBILITY STATEMENT

We are committed to the reproducibility of our work. Our results are verifiable through the following resources:

- **Source Code:** The complete source code, including implementations of our proposed stretch transformations, all baseline methods, and the experimental evaluation pipeline, is provided anonymously in the supplementary material.
- **Hyperparameters:** A comprehensive description of the experimental setup, including the full hyperparameter search spaces for all models and transformations, is detailed in Appendix C.3.
- **Theoretical Derivations:** All theoretical derivations for both the unsupervised and supervised stretch transformations are provided in full detail in Appendix A.
- **Datasets:** All datasets used in our experiments are from the publicly available TALENT benchmark, and details on their selection are provided in Appendix C.3.

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

## A  THEORETICAL FOUNDATIONS AND DERIVATIONS

### A.1  GENERAL OBJECTIVE: FROM KERNEL SMOOTHNESS TO DIRICHLET ENERGY

**Motivation via Heat Kernel Expansion.**  The primary goal is to find a transformation that makes the target function smoother in the transformed space. We quantify smoothness using kernel correlation with a Gaussian RBF kernel, $k_\sigma(y, z) = \exp(-(y - z)^2/(2\sigma^2))$. The integral operator $T_\sigma$ associated with this kernel can be expressed via the heat kernel expansion as $T_\sigma = e^{\frac{\sigma^2}{2}\partial_{yy}}$, where $\partial_{yy}$ is the second derivative operator (Varadhan, 1967; Molchanov, 1975). For a small bandwidth $\sigma$, a Taylor expansion gives $T_\sigma = I + \frac{\sigma^2}{2}\partial_{yy} + O(\sigma^4)$.

To isolate the smoothness-dependent term, we consider the centered kernel correlation:

$$\langle g, (T_\sigma - I)g \rangle = \left\langle g, \left(\frac{\sigma^2}{2}\partial_{yy} + O(\sigma^4)\right) g \right\rangle = \frac{\sigma^2}{2}\int_0^1 g(y) \cdot g''(y)\, dy + O(\sigma^4). \quad (18)$$

Using integration by parts (assuming Neumann boundary conditions, $g'(0) = g'(1) = 0$), this simplifies to:

$$\int_0^1 g(y) \cdot g''(y)\, dy = [g(y) \cdot g'(y)]_0^1 - \int_0^1 g'(y) \cdot g'(y)\, dy = -\int_0^1 \|g'(y)\|_2^2\, dy. \quad (19)$$

Therefore, maximizing kernel smoothness in this regime is equivalent to minimizing the Dirichlet energy:

$$\langle g, (T_\sigma - I)g \rangle = -\frac{\sigma^2}{2}\int_0^1 \|g'(y)\|_2^2\, dy + O(\sigma^4), \quad \text{so we minimize } \mathcal{E}[g] = \int_0^1 \|g'(y)\|_2^2\, dy. \quad (20)$$

**Discretization of Dirichlet Energy.**  For a dataset $\{x_i\}_{i=1}^n$ (assumed sorted) with transformed values $y_i = s(x_i)$, we approximate the continuous target function $g(y)$ by its piecewise linear interpolant, denoted by $\tilde{g}$. The interpolant $\tilde{g}$ is constructed by connecting the points $(y_i, f(x_i))$. By the definition of our transformation, the true function value at the transformed point $y_i$ is precisely $g(y_i) = f(s^{-1}(y_i)) = f(x_i)$.

The derivative of this interpolant, $\tilde{g}'(y)$, is constant over any interval $[y_i, y_{i+1}]$ and is given by:

$$\tilde{g}'(y) = \frac{f(x_{i+1}) - f(x_i)}{y_{i+1} - y_i} = \frac{\Delta f_i}{\Delta y_i}$$

The total discrete Dirichlet energy is the integral of the squared norm of this derivative, which simplifies to a sum over the intervals:

$$\mathcal{E}_{\text{disc}} = \sum_{i=1}^{n-1} \int_{y_i}^{y_{i+1}} \|\tilde{g}'(y)\|_2^2\, dy = \sum_{i=1}^{n-1} \frac{\|\Delta f_i\|_2^2}{\Delta y_i}. \quad (21)$$

This discrete energy serves as the foundation for both our unsupervised and supervised methods.

### A.2  UNSUPERVISED STRETCH: DERIVATION OF THE MINIMAX OBJECTIVE

**Worst-Case Dirichlet Energy Formulation.**  Without target information, we formulate a minimax problem to find the spacing allocation that minimizes the worst-case Dirichlet energy. The optimization is performed over all possible target functions $f$ with a bounded total variation. Since the discrete Dirichlet energy $\mathcal{E}_{\text{disc}} = \sum_{i=1}^{n-1} \frac{\|\Delta f_i\|_2^2}{\Delta y_i}$ depends only on the sequence of difference vectors $\{\Delta f_i\}_{i=1}^{n-1}$, this is equivalent to solving the following problem:

$$\max_{\{\Delta f_i\}} \sum_{i=1}^{n-1} \frac{\|\Delta f_i\|_2^2}{\Delta y_i} \quad \text{s.t.} \quad \|\Delta f_i\|_2^2 \le C, \ \forall i. \quad (22)$$

This bounded local variation constraint assumes that for an unknown function, the magnitude of its change over any small interval is bounded, which is a practical assumption for unsupervised

scenarios where we want to be robust against arbitrary local fluctuations. Therefore, this problem can be simplified by optimizing directly over these scalar norm values, which justifies the subsequent step of taking the derivative of the Lagrangian with respect to $\|\Delta f_i\|_2$.

**Solving for Worst-Case Target Function.** For a fixed spacing allocation $\{\Delta y_i\}$, the worst-case energy is found by maximizing $\sum_{i=1}^{n-1} \frac{\|\Delta f_i\|_2^2}{\Delta y_i}$ subject to $\|\Delta f_k\|_2^2 \leq C$ for all $k$. Since each constraint is independent, the solution is straightforward: to maximize the objective, we set each $\|\Delta f_i\|_2^2 = C$. Substituting this back into the energy formula, the worst-case energy becomes:

$$\mathcal{E}_{\text{worst}}(\{\Delta y_i\}) = \sum_{i=1}^{n-1} \frac{C}{\Delta y_i} = C \sum_{i=1}^{n-1} \frac{1}{\Delta y_i}. \tag{23}$$

Our goal is to find the spacing allocation $\{\Delta y_i\}$ that minimizes this worst-case energy. This leads directly to the optimization problem:

$$\min_{\{\Delta y_i > 0\}} \sum_{i=1}^{n-1} \frac{1}{\Delta y_i} \quad \text{s.t.} \quad \sum_{i=1}^{n-1} \Delta y_i = 1. \tag{24}$$

**Optimal Spacing Allocation.** The constrained optimization problem is solved using the method of Lagrange multipliers. We form the Lagrangian

$$\mathcal{L} = \sum_{i=1}^{n-1} \frac{1}{\Delta y_i} - \mu \left( \sum_{i=1}^{n-1} \Delta y_i - 1 \right). \tag{25}$$

Setting the partial derivative with respect to an arbitrary $\Delta y_k$ to zero gives the first-order condition:

$$\frac{\partial \mathcal{L}}{\partial \Delta y_k} = -\frac{1}{(\Delta y_k)^2} - \mu = 0. \tag{26}$$

This condition implies that $(\Delta y_k)^2 = -1/\mu$, which must hold for all $k \in \{1, \ldots, n-1\}$. Therefore, all spacings $\Delta y_k$ must be equal. Enforcing the constraint $\sum_{i=1}^{n-1} \Delta y_i = 1$ leads to the unique solution of a uniform spacing:

$$\Delta y_i^* = \frac{1}{n-1} \quad \text{for all } i. \tag{27}$$

**Application to Binning Framework.** In our stretch transformation with $T$ quantile bins, each bin $t$ contains $n_t \approx n/T$ samples. Applying the uniform spacing principle within bin $t$ with allocated width $w_t$, the contribution to the objective function from bin $t$ is $(n_t - 1)/w_t$. The global optimization over bin widths becomes:

$$\min_{\{w_t > 0\}} \sum_{t=1}^{T} \frac{n_t - 1}{w_t} \quad \text{s.t.} \quad \sum_{t=1}^{T} w_t = 1. \tag{28}$$

The solution, obtained via Lagrange multipliers, is $w_t \propto \sqrt{n_t - 1}$. Under quantile binning, where $n_t$ is approximately constant, this simplifies to a uniform width allocation:

$$w_t^* = \frac{1}{T} \quad \text{for all } t = 1, \ldots, T. \tag{29}$$

### A.3 SUPERVISED STRETCH: DERIVATION OF THE TARGET-INFORMED OBJECTIVE

**The Exact Objective and its Instability** A direct approach to minimizing the discrete Dirichlet energy $\mathcal{E}_{\text{disc}}$ with our piecewise linear map reveals a potential instability issue in practice. Within bin $t$, the slope $s'(x) = w_t/(b_t - b_{t-1})$ is constant. Therefore, the transformed spacing is proportional

to the original spacing, that is, $\Delta y_i = \frac{w_t}{b_t - b_{t-1}} \Delta x_i$. Substituting this into the energy formula, the contribution from bin $t$ becomes:

$$\mathcal{E}_t = \sum_{i \in \mathcal{I}_t} \frac{\|\Delta f_i\|_2^2}{\Delta y_i} = \frac{b_t - b_{t-1}}{w_t} \sum_{i \in \mathcal{I}_t} \frac{\|\Delta f_i\|_2^2}{\Delta x_i}. \tag{30}$$

Minimizing the total energy involves terms of the form $\|\Delta f_i\|_2^2 / \Delta x_i$. This expression though can be highly unstable. For instance, if a region is densely sampled ($\Delta x_i \to 0$), even infinitesimal noise in $f$ can cause this term to diverge. An objective based on this quantity would be pathologically sensitive to sample density and noise.

**A Robust Alternative: Optimizing a Lower Bound**  To overcome this instability, we formulate a more robust objective. Instead of enforcing the rigid link between $\Delta y_i$ and $\Delta x_i$, we relax the problem. For a bin $t$, we fix its total width $w_t = \sum_{i \in \mathcal{I}_t} \Delta y_i$ and find the minimum possible energy contribution by optimizing over all possible internal spacings $\{\Delta y_i > 0\}_{i \in \mathcal{I}_t}$.

We find this minimum using the Cauchy-Schwarz inequality. Let $a_i = \|\Delta f_i\|_2$ and $b_i = \sqrt{\Delta y_i}$. Then:

$$\left( \sum_{i \in \mathcal{I}_t} \|\Delta f_i\|_2 \right)^2 = \left( \sum_{i \in \mathcal{I}_t} \frac{\|\Delta f_i\|_2}{\sqrt{\Delta y_i}} \cdot \sqrt{\Delta y_i} \right)^2 \leq \left( \sum_{i \in \mathcal{I}_t} \frac{\|\Delta f_i\|_2^2}{\Delta y_i} \right) \left( \sum_{i \in \mathcal{I}_t} \Delta y_i \right). \tag{31}$$

Rearranging gives the desired tight lower bound on the bin's energy contribution:

$$\sum_{i \in \mathcal{I}_t} \frac{\|\Delta f_i\|_2^2}{\Delta y_i} \geq \frac{\left( \sum_{i \in \mathcal{I}_t} \|\Delta f_i\|_2 \right)^2}{w_t} = \frac{S_t^2}{w_t}, \tag{32}$$

where $S_t := \sum_{i \in \mathcal{I}_t} \|\Delta f_i\|_2$ is the total target variation in bin $t$. The quantity $S_t$ is a stable, aggregated measure, robust to the issues of individual noisy or dense samples.

**Optimal Bin Width Allocation**  We adopt this robust lower bound as our objective and find the optimal widths $\{w_t\}$ by solving:

$$\min_{\{w_t > 0\}} \sum_{t=1}^{T} \frac{S_t^2}{w_t} \quad \text{s.t.} \quad \sum_{t=1}^{T} w_t = 1. \tag{33}$$

The Lagrangian for this convex problem is $\mathcal{L}(\{w_t\}, \lambda) = \sum_{t=1}^{T} \frac{S_t^2}{w_t} - \lambda(\sum_{t=1}^{T} w_t - 1)$. Setting the partial derivatives to zero yields $\frac{\partial \mathcal{L}}{\partial w_t} = -\frac{S_t^2}{w_t^2} - \lambda = 0$, which implies $w_t \propto S_t$. Substituting this into the constraint $\sum w_t = 1$ gives the closed-form solution:

$$w_t^* = \frac{S_t}{\sum_{u=1}^{T} S_u}. \tag{34}$$

The optimal width for each bin is therefore proportional to the total target variation within it.

# B   DETAILED PAIRWISE PERFORMANCE ANALYSIS

To provide a more granular analysis of the stretch transformation's performance, this section extends the summary presented in the win-rate heatmap Section 4 (Figure 3). Recall that our pairwise comparison uses a stringent win-loss-tie (W-L-T) criterion, where a "win" is declared only if a method's performance exceeds another's by more than the larger of their respective standard errors. The win-rate percentages shown in the heatmap are summary statistics derived directly from the raw W-L-T counts across all 190 dataset-model combinations.

Here, we look closer at these raw counts. Figures 4 through 9 present the detailed scatter plots comparing our supervised and unsupervised methods against three key baselines: PLE, PLE-T, and standardization. In each figure, every point represents a single dataset-model combination. Points located above the diagonal indicate a win for the stretch transformation, points below indicate a loss, and points on the diagonal represent a statistical tie. The total win-loss-tie counts are summarized and displayed directly on each plot, providing the quantitative confirmation for the performance advantages discussed in Section 4.2.

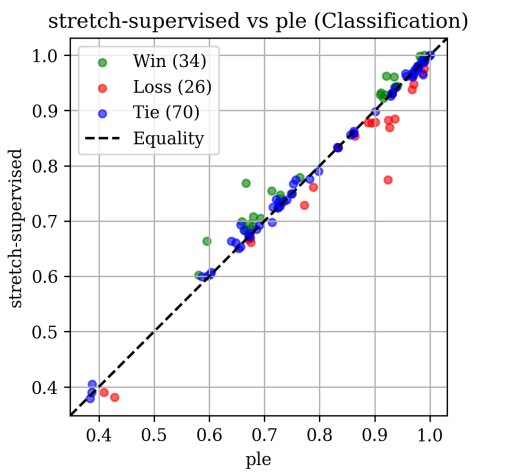 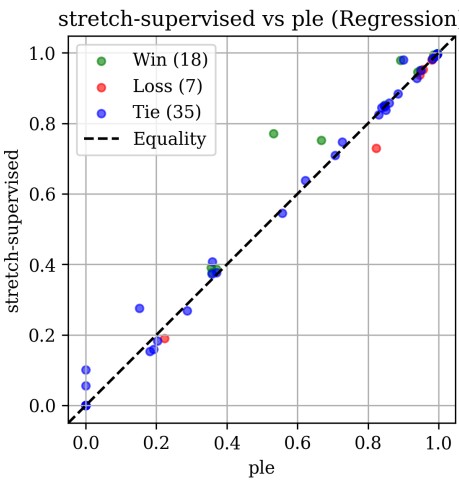

Figure 4: Stretch Supervised vs PLE

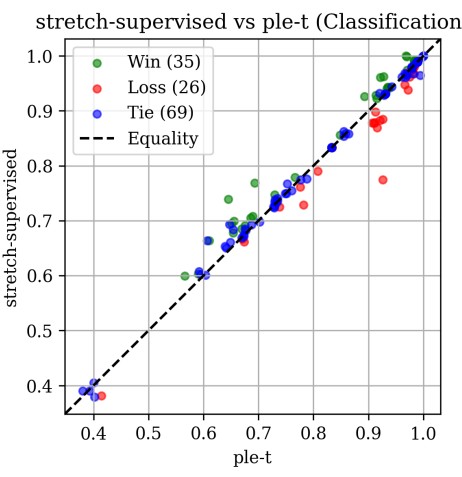 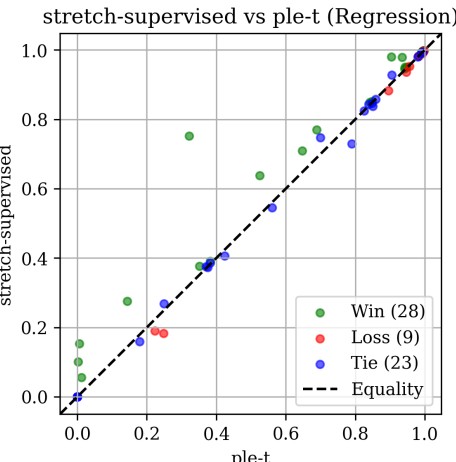

Figure 5: Stretch Supervised vs PLE-T

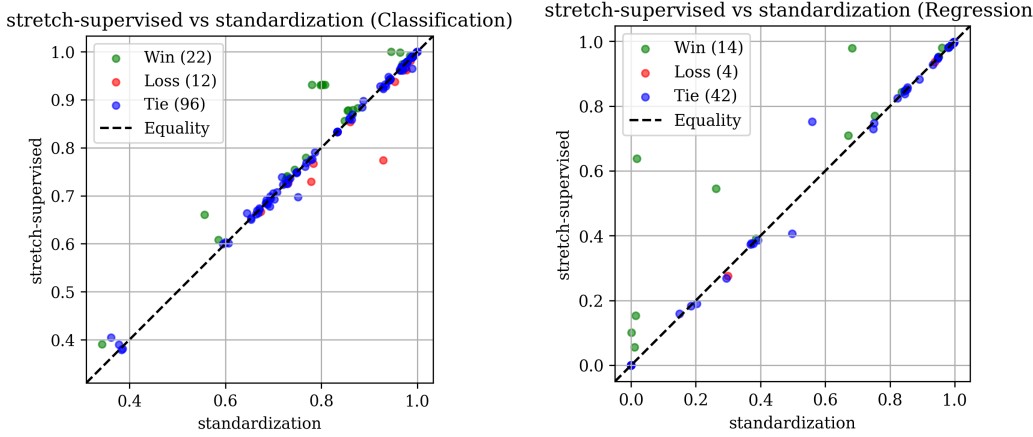

Figure 6: Stretch Supervised vs Standardization

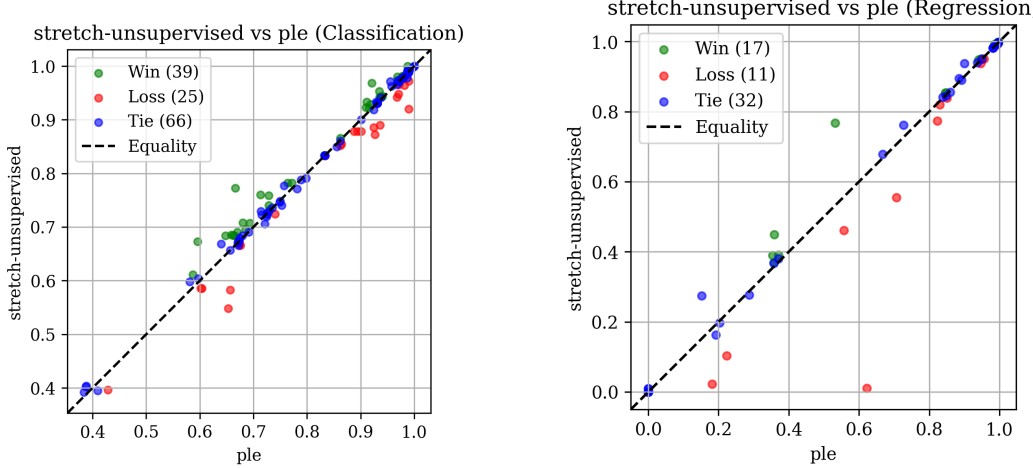

Figure 7: Stretch Unsupervised vs PLE

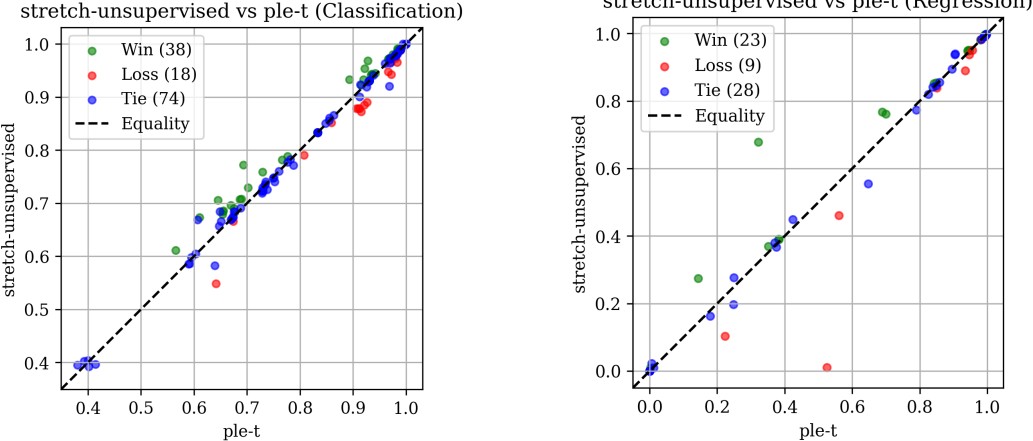

Figure 8: Stretch Unsupervised vs PLE-T

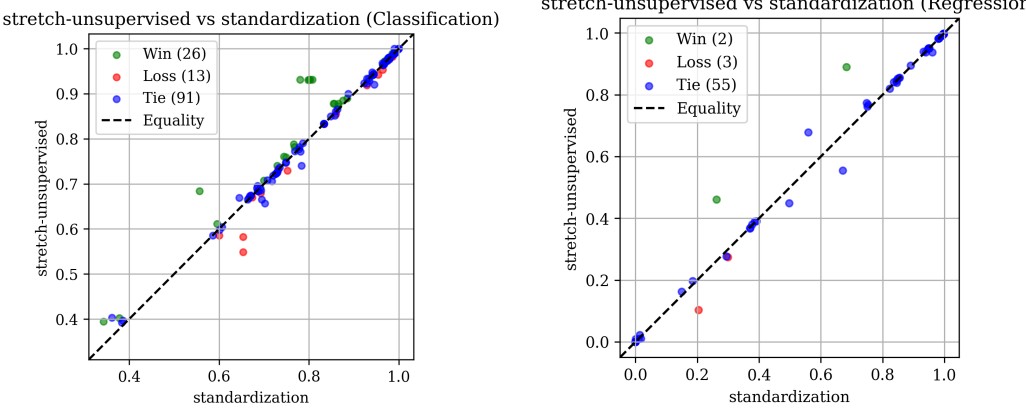

Figure 9: Stretch Unsupervised vs Standardization

## C  IMPLEMENTATION AND EXPERIMENTAL DETAILS

### C.1  OUT-OF-FOLD KERNEL REGRESSION FOR SUPERVISED STRETCH

A naive implementation of supervised stretch would use the entire training set to estimate the target function $f(x)$ needed to compute the bin widths $\{w_t\}$. This, however, would introduce significant information leakage. The transformation $s$ would be designed using the same target values $t_i$ that the downstream model $g_\theta$ is later trained to predict. In essence, the target information would be used twice, once to shape the feature space, and again to train the model within that space. This can lead to overly optimistic performance estimates and poor generalization.

To prevent this, we employ a standard K-fold cross-validation scheme to obtain out-of-fold (OOF) estimates of the marginal regression function, $\widehat{f}(x_i)$, for each sample. This ensures that the target value $t_i$ is never used in the process of creating the transformed feature $s(x_i)$ that it will later be paired with during model training.

For classification, we use stratified K-fold ($K = 10$, or fewer if a class is too small) on the one-hot encoded targets. For regression, we use standard K-fold. Within each fold, we fit an adaptive Nadaraya-Watson kernel regressor on the training portion of the data to generate predictions for the validation portion (Nadaraya, 1964; Watson, 1964). Let $\mathcal{D}_{\text{tr}}$ denote the set of indices of the samples in the training portion for a given fold. The conditional expectation at a point $x$ in the validation set is then estimated as:

$$\widehat{f}(x) = \frac{\sum_{j \in \mathcal{D}_{\text{tr}}} K_h(x, x_j) \cdot Y_j}{\sum_{j \in \mathcal{D}_{\text{tr}}} K_h(x, x_j) + \epsilon}, \quad \text{where} \quad K_h(x, x_j) = \exp\left(-\frac{(x - x_j)^2}{2h(x)^2}\right). \tag{35}$$

Here, $Y_j$ is the target value for the $j$-th sample, and the adaptive bandwidth $h(x)$ is determined by the distance to the $k$-th nearest neighbor of $x$ within the training data, allowing the smoothness of the estimate to vary with local sample density.

After obtaining OOF predictions for all samples, we sort them by the feature value and compute the total variation $S_t$ of these predictions within each quantile bin. Several fallback mechanisms ensure robustness: if a feature has too few unique values, or if the total target variation is negligible, we revert to simpler transformations (identity or unsupervised stretch).

### C.2  COMPUTATIONAL FALLBACK SCHEME FOR PLE

PLE expands feature dimensionality from $d$ to $d \times T$, which can become computationally prohibitive. In our experiments, if the resulting tensor size exceeds a memory limit, we use a fallback scheme that substitutes standardization for PLE on that specific dataset-model combination. This affects fewer than 5% of the 190 combinations and does not impact our proposed methods.

## C.3 Experimental Setup Details

**Dataset Selection.** From the original TALENT Benchmark 1 collection of 42 datasets, we exclude four datasets that contain only categorical features (Amazon_employee_access, BNG(tic-tac-toe), led24, splice), yielding our final experimental suite of 38 datasets.

**Categorical Feature Handling.** We use indices encoding as the default. FT-Transformer learns embeddings, RealMLP internally converts to one-hot, and other models use the indices directly. For feature-expanding transformations like PLE, FT-Transformer applies a linear projection to each expanded dimension to create tokens.

**Hyperparameter Search Spaces.** We conduct Bayesian optimization using Optuna with 100 trials per configuration. The search spaces for models and transformations are detailed in Table 3 and Table 2. We follow the hyperparameter search space design from the original TALENT benchmark (Ye et al., 2024; Liu et al., 2024), including their special syntax for sampling methods, which we define here for clarity:

- `?distribution[default, ...args]`: An "optional" parameter. With 50% probability, the `default` value is used (e.g., 0.0 for dropout, disabling it). Otherwise, a value is sampled from the specified `distribution` using the remaining arguments.
- `$name[...]`: A special, complex sampling function defined by the benchmark's codebase. For instance, `$mlp_d_layers` samples the number of layers and then the width of each layer within given bounds, with different sampling strategies for the first, middle, and last layers.

Table 2: Transformation hyperparameter search spaces.

| Transformation | Parameter Search Space |
|---|---|
| Supervised/Unsupervised Stretch | n_bins: `?categorical[1, 2, 4, 8, 16, 32, 64, 128, 256, 512, 100000]`[†] |
| PLE | n_bins: `int[2, 256]` |
| Other transformations | No hyperparameters |

Table 3: Model hyperparameter search spaces.

| Model | Parameter | Search Space |
|---|---|---|
| MLP | d_layers | `$mlp_d_layers[1, 8, 64, 2048]`[*] |
| | dropout | `?uniform[0.0, 0.0, 0.5]`[†] |
| | lr | `loguniform[1e-5, 0.01]` |
| | weight_decay | `?loguniform[0.0, 1e-6, 0.001]`[†] |
| MLP-PLR | d_layers | `$mlp_d_layers[1, 8, 64, 1024]`[*] |
| | dropout | `?uniform[0.0, 0.0, 0.5]`[†] |
| | n_frequencies | `int[16, 96]` |
| | frequency_scale | `loguniform[0.01, 100.0]` |
| | d_embedding | `int[16, 64]` |
| | lr | `loguniform[1e-5, 0.01]` |
| | weight_decay | `?loguniform[0.0, 1e-6, 0.001]`[†] |
| FT-Transformer | n_layers | `int[1, 4]` |
| | d_token | `categorical[8, 16, 32, 64, 128]` |
| | residual_dropout | `?uniform[0.0, 0.0, 0.2]`[†] |
| | attention_dropout | `uniform[0.0, 0.5]` |
| | ffn_dropout | `uniform[0.0, 0.5]` |
| | d_ffn_factor | `uniform[0.67, 2.67]` |
| | lr | `loguniform[1e-5, 0.001]` |
| | weight_decay | `loguniform[1e-6, 0.001]` |
| RealMLP | num_emb_type | `categorical[none, pbld, pl, plr]` |
| | add_front_scale | `categorical[true, false]` |
| | hidden_sizes | `categorical[[256,256,256], [64,64,64,64,64], [512]]` |
| | p_drop | `categorical[0.0, 0.15, 0.30]` |
| | act | `categorical[selu, relu, mish]` |
| | lr | `loguniform[0.02, 0.3]` |
| | wd | `categorical[0.0, 0.02]` |
| | ls_eps | `categorical[0.0, 0.1]` |
| | plr_sigma | `loguniform[0.05, 0.5]` |
| ResNet | n_layers | `int[1, 8]` |
| | d | `int[64, 512]` |
| | d_hidden_factor | `uniform[1.0, 4.0]` |
| | hidden_dropout | `uniform[0.0, 0.5]` |
| | residual_dropout | `?uniform[0.0, 0.0, 0.5]`[†] |
| | lr | `loguniform[1e-5, 0.01]` |
| | weight_decay | `?loguniform[0.0, 1e-6, 0.001]`[†] |

# D DATASET COMPOSITION AND FEATURE TYPE CHARACTERISTIC

To demonstrate the applicability of our method across datasets with varying feature type compositions, we provide a systematic analysis of our benchmark's characteristics. We categorize the 38 datasets into three groups based on the relative proportion of numeric versus categorical features:

- **Category Dominant:** Datasets where categorical features significantly outnumber numeric features (typically a ratio of 3:1 or greater). This group contains 4 datasets including highly categorical datasets.
- **Balanced:** Datasets where numeric and categorical features are roughly comparable in number (within a 2:1 ratio in either direction). This group contains 8 datasets with diverse feature compositions.
- **Numeric Dominant:** Datasets where numeric features significantly outnumber categorical features or contain only numeric features. This group contains 26 datasets, ranging from moderately numeric-dominant to purely numeric.

Table 4 presents detailed statistics for each dataset, including sample sizes ranging from approximately 650 to over 65,000 samples, and feature counts spanning from a handful to over 140 features. The imbalance ratio column indicates class imbalance for classification tasks where applicable.

The diverse composition of our benchmark ensures that our evaluation is not biased toward numeric-heavy datasets. Our experimental results show that supervised stretch achieves the best overall performance when evaluated across the full benchmark, while maintaining strong competitiveness across all three feature type categories. The robustness across datasets with varying sample sizes and dimensionalities further supports the practical applicability of our framework.

**Relevance to Performance Analysis**: Understanding this composition is critical for evaluating the generality of numeric feature transformations. A common critique is that numeric transformations may lose relevance in mixed-type tabular data. To investigate this, we utilize these three categories to stratify our performance analysis. The detailed performance scores for each group are reported in Appendix E (Table 5). This grouping allows us to explicitly verify whether the benefits of Stretch transformation persist when numeric features are not the sole source of information.

Table 4: Dataset characteristics grouped by feature type dominance.

| Dataset | Num Features | Cat Features | Total Samples | Imbalance Ratio | N Classes |
|---|---|---|---|---|---|
| **Balanced Datasets** | | | | | |
| Diamonds | 6 | 3 | 53940 | – | – |
| E-CommereShippingData | 6 | 4 | 10999 | – | – |
| Fitness_Club_c | 3 | 3 | 1500 | – | – |
| Kaggle_bike_sharing_demand | 3 | 6 | 10886 | – | 1 |
| NHANES_age_prediction | 4 | 3 | 2277 | – | – |
| Shop_Customer_Data | 4 | 2 | 2000 | – | – |
| compass | 8 | 9 | 16644 | 1.000 | – |
| estimation_of_obesity_levels | 8 | 8 | 2111 | – | – |
| **Category Dominant Datasets** | | | | | |
| archive_r56_Portuguese | 1 | 29 | 651 | – | – |
| okcupid_stem | 2 | 11 | 26677 | 6.826 | 3 |
| socmob | 1 | 4 | 1156 | – | – |
| thyroid-dis | 6 | 20 | 2800 | 52.645 | 5 |
| **Numeric Dominant Datasets** | | | | | |
| ASP-POTASSCO-classification | 140 | 1 | 1294 | 12.286 | 11 |
| Ailerons | 40 | 0 | 13750 | – | – |
| Bias_correction_r_2 | 21 | 0 | 7725 | – | – |
| Click_prediction_small | 3 | 0 | 39948 | – | – |
| Contaminant-detection-11.0GHz | 30 | 0 | 2400 | – | – |
| FOREX_audjpy-day-High | 10 | 0 | 1832 | 1.056 | 2 |
| FOREX_audsgd-hour-High | 10 | 0 | 43825 | 1.061 | 2 |
| IEEE80211aa-GATS | 27 | 0 | 4046 | – | – |
| Intersectional-Bias-Assessment | 14 | 5 | 11000 | – | 2 |
| Job_Profitability | 27 | 1 | 14480 | – | – |
| Mobile_Price_Classification | 14 | 6 | 2000 | – | – |
| VulNoneVul | 16 | 0 | 5692 | – | 2 |
| Waterstress | 22 | 0 | 1188 | – | 2 |
| electricity | 7 | 1 | 45312 | 1.355 | 2 |
| htru | 8 | 0 | 17898 | – | – |
| ibm-employee-performance | 23 | 7 | 1470 | 5.504 | 2 |
| internet_firewall | 7 | 0 | 65532 | – | – |
| jungle_chess_2pcs_endgame | 6 | 0 | 44819 | 5.320 | 3 |
| kc1 | 21 | 0 | 2109 | 5.469 | 2 |
| optdigits | 64 | 0 | 5620 | 1.032 | 10 |
| page-blocks | 10 | 0 | 5473 | 175.464 | 5 |
| pol | 26 | 0 | 10082 | 1.000 | 2 |
| pol_reg | 48 | 0 | 15000 | – | – |
| pole | 26 | 0 | 14998 | – | – |
| wine | 4 | 0 | 2554 | 1.000 | 2 |
| yeast | 8 | 0 | 1484 | 92.600 | 10 |

# E  PERFORMANCE BREAKDOWN BY FEATURE COMPOSITION

To understand how feature heterogeneity affects transformation efficacy, Table 5 breaks down the model performance based on the dataset categories defined in Appendix D (Numeric Dominant, Balanced, and Category Dominant).

- **Numeric Dominant**: As anticipated, Supervised Stretch consistently leads in datasets where numeric features constitute the majority, confirming the direct benefit of our optimization target.
- **Category Dominant**: Even when categorical features significantly outnumber numeric features, the Stretch framework remains competitive, confirming the broad applicability of our method. Specifically, Supervised Stretch achieves the highest Overall Score in regression, and Unsupervised Stretch achieves the highest Overall Score in classification within this category. Our results show that optimizing the representation of even a minority of numeric features yields significant gains, demonstrating the method's value in mixed-type tabular data.

Table 5: Performance by dataset category and task type (average score).

| Category | Task | Model | Sup. | Unsup. | Minmax | Ple | Ple T | Quantile | RS-SC | Std | YJ |
|---|---|---|---|---|---|---|---|---|---|---|---|
| Balanced | Classification | ftt | 0.7987 | 0.8000 | 0.7911 | 0.7967 | 0.7994 | **0.8023** | 0.8005 | 0.7987 | 0.8014 |
| | | mlp | 0.7673 | 0.7827 | 0.7738 | 0.7860 | **0.7907** | 0.7809 | 0.7855 | 0.7813 | 0.7849 |
| | | mlp_plr | 0.7843 | **0.7994** | 0.7850 | 0.7891 | 0.7880 | 0.7875 | 0.7926 | 0.7916 | 0.7932 |
| | | realmlp | 0.8008 | 0.8020 | 0.7984 | 0.7990 | **0.8101** | 0.8021 | 0.8001 | 0.8005 | 0.8009 |
| | | resnet | 0.7844 | 0.7793 | 0.7735 | 0.7777 | **0.7871** | 0.7795 | 0.7817 | 0.7834 | 0.7868 |
| | | **Overall** | 0.7871 | 0.7927 | 0.7844 | 0.7897 | **0.7950** | 0.7905 | 0.7921 | 0.7911 | 0.7934 |
| | Regression | ftt | 0.5493 | 0.5478 | 0.5422 | 0.5418 | 0.5477 | 0.5482 | **0.5503** | 0.5479 | 0.5503 |
| | | mlp | 0.4404 | 0.4300 | 0.4740 | 0.4051 | 0.4321 | 0.4747 | 0.4736 | 0.3896 | **0.4961** |
| | | mlp_plr | 0.5340 | 0.5346 | **0.5375** | 0.4708 | 0.5128 | 0.5313 | 0.5346 | 0.5311 | 0.5327 |
| | | realmlp | **0.5501** | 0.5475 | 0.5357 | 0.5445 | 0.5491 | 0.5419 | 0.5478 | 0.5471 | 0.5435 |
| | | resnet | 0.5259 | 0.5172 | 0.5225 | 0.4962 | 0.4889 | 0.5274 | **0.5326** | 0.5207 | 0.5254 |
| | | **Overall** | 0.5199 | 0.5154 | 0.5224 | 0.4917 | 0.5061 | 0.5247 | 0.5278 | 0.5073 | **0.5296** |
| Category Dominant | Classification | ftt | **0.7207** | 0.7196 | 0.7185 | 0.7206 | 0.7191 | 0.7161 | 0.7199 | 0.7205 | 0.7119 |
| | | mlp | 0.7063 | **0.7125** | 0.7054 | 0.7072 | 0.6993 | 0.7030 | 0.7102 | 0.7085 | 0.7077 |
| | | mlp_plr | 0.7076 | 0.7086 | 0.7093 | 0.6978 | 0.7046 | 0.7082 | **0.7128** | 0.7098 | 0.7093 |
| | | realmlp | **0.7202** | 0.7154 | 0.7168 | 0.7141 | 0.7122 | 0.7160 | 0.7173 | 0.7169 | 0.7160 |
| | | resnet | 0.7107 | 0.7133 | **0.7171** | 0.7025 | 0.7049 | 0.7091 | 0.7077 | 0.7094 | 0.7116 |
| | | **Overall** | 0.7131 | **0.7139** | 0.7134 | 0.7084 | 0.7080 | 0.7105 | 0.7136 | 0.7130 | 0.7113 |
| | Regression | ftt | 0.5797 | 0.5845 | 0.5241 | 0.5178 | 0.5191 | 0.5790 | 0.5374 | **0.5951** | 0.5857 |
| | | mlp | **0.4713** | 0.3908 | 0.4261 | 0.4453 | 0.2726 | 0.4377 | 0.3772 | 0.3813 | 0.4507 |
| | | mlp_plr | 0.4564 | 0.4850 | 0.4760 | 0.5130 | **0.5190** | 0.4576 | 0.4366 | 0.4666 | 0.4646 |
| | | realmlp | 0.5984 | 0.6088 | 0.5979 | 0.6122 | 0.5774 | **0.6251** | 0.5889 | 0.6134 | 0.5854 |
| | | resnet | 0.4342 | 0.3587 | **0.4895** | 0.4493 | 0.4134 | 0.3550 | 0.3818 | 0.4104 | 0.3532 |
| | | **Overall** | **0.5080** | 0.4856 | 0.5027 | 0.5076 | 0.4603 | 0.4909 | 0.4644 | 0.4934 | 0.4879 |
| Numeric Dominant | Classification | ftt | **0.8340** | 0.8287 | 0.8048 | 0.8315 | 0.8305 | 0.8331 | 0.8216 | 0.8179 | 0.8252 |
| | | mlp | 0.8313 | 0.8326 | 0.8234 | **0.8362** | 0.8283 | 0.8270 | 0.8306 | 0.8254 | 0.8294 |
| | | mlp_plr | 0.8331 | 0.8309 | 0.8269 | 0.8279 | 0.8273 | **0.8348** | 0.8278 | 0.8250 | 0.8255 |
| | | realmlp | 0.8444 | 0.8451 | **0.8477** | 0.8401 | 0.8392 | 0.8450 | 0.8459 | 0.8379 | 0.8401 |
| | | resnet | **0.8394** | 0.8374 | 0.8248 | 0.8366 | 0.8339 | 0.8296 | 0.8387 | 0.8301 | 0.8307 |
| | | **Overall** | **0.8364** | 0.8349 | 0.8255 | 0.8345 | 0.8318 | 0.8339 | 0.8329 | 0.8273 | 0.8302 |
| | Regression | ftt | 0.8228 | 0.8017 | 0.7964 | **0.8260** | 0.7961 | 0.8041 | 0.7991 | 0.7993 | 0.7716 |
| | | mlp | **0.8100** | 0.7951 | 0.7977 | 0.7969 | 0.7964 | 0.7757 | 0.7969 | 0.7950 | 0.6096 |
| | | mlp_plr | 0.8884 | 0.8746 | 0.8011 | 0.8868 | **0.8892** | 0.8310 | 0.8600 | 0.8413 | 0.7879 |
| | | realmlp | **0.9063** | 0.8013 | 0.8016 | 0.9043 | 0.8884 | 0.8435 | 0.8162 | 0.8023 | 0.8089 |
| | | resnet | **0.8060** | 0.7980 | 0.7979 | 0.7966 | 0.7977 | 0.7808 | 0.7974 | 0.7979 | 0.6162 |
| | | **Overall** | **0.8467** | 0.8141 | 0.7990 | 0.8421 | 0.8335 | 0.8070 | 0.8139 | 0.8072 | 0.7188 |

# F  MARGINAL SIGNAL NON-UNIFORMITY ANALYSIS

Our method is primarily designed for numeric features with sharp variation. If features with primarily flat or weak marginal relationships predominate in tabular datasets, our method would have limited applicability. To address this concern, we conduct a systematic quantitative analysis of marginal signal structures across all datasets in our benchmark. Contrary to what might be intuitive, our analysis demonstrates that non-uniform marginal patterns, characterized by localized regions of sharp variation, are indeed the common case in real-world tabular data.

## F.1  METHODOLOGY

For each numeric feature across all benchmark datasets, we employ the following procedure to quantify the non-uniformity of its marginal effect on the target:

**Step 1: Data Preprocessing.** For all numeric features, we impute missing values with the mean and standardize to zero mean and unit variance for cross-dataset comparison.

**Step 2: Marginal Function Construction.** For each unique feature value $x$, we construct the marginal expectation $f(x) = \mathbb{E}[t|x]$ following the same approach used throughout our framework. For regression, this is the mean of standardized continuous targets. For classification, this expectation represents the empirical class probability distribution $f(x) \in \mathbb{R}^K$, consistent with our method's treatment of vector-valued targets (see footnote 1).

**Step 3: Local Marginal Slope Computation.** For consecutive unique feature values $x_i$ and $x_{i+1}$, we compute $\Delta f_i = f(x_{i+1}) - f(x_i)$. We then calculate the local marginal slope magnitude $|\Delta f_i/\Delta x_i|$ where $|\cdot|$ denotes absolute value for regression or L2 norm for classification vectors.

**Step 4: Relative Marginal Slope Normalization.** To isolate the internal variation pattern from overall signal strength, we compute:

$$\text{Relative Marginal Slope}_i = \frac{|\Delta f_i/\Delta x_i| \cdot n}{\sum_{j=1}^{n} |\Delta f_j/\Delta x_j|} \tag{36}$$

where $n$ is the number of local slopes for that feature. This normalization ensures each feature's distribution has mean 1. Under perfect uniformity (constant rate of change), all values would equal 1. Non-uniform patterns produce dispersion: regions with relatively rapid changes yield values greater than 1, while relatively flat regions yield values near 0.

**Step 5: Dataset-Level Aggregation.** We pool all relative marginal slopes from all numeric features within each dataset to form a single distribution per dataset.

**Step 6: Distribution Metrics.** We compute the standard deviation and skewness of each dataset's pooled distribution. Standard deviation quantifies overall variability in marginal slopes, while positive skewness indicates the presence of localized regions with exceptionally large slopes.

## F.2  RESULTS AND INTERPRETATION

Table 6 presents the metrics for all datasets with task type annotations. Our analysis reveals several key findings. Most notably, non-uniform marginal patterns are the norm across the tabular datasets in our benchmark.

**Non-Uniform Marginal Patterns Are the Norm, Not the Exception.** Tabular datasets exhibit substantial standard deviations, ranging from 0.47 to 20.15 with a median of 1.1. This indicates significant variation in marginal slopes across features and regions. If features predominantly had flat marginal relationships with targets, we would observe values concentrated near zero with minimal standard deviation. Instead, the widespread substantial variation demonstrates that features typically exhibit meaningful marginal effects with considerable heterogeneity across different regions of the feature space.

Furthermore, most datasets show positive skewness, indicating the presence of localized regions where marginal slopes are exceptionally large. This pervasive pattern validates that non-uniform marginal effects, characterized by mixing regions of different gradient magnitudes, is a common characteristic of real-world tabular data rather than an exception. This directly confirms the scenario our supervised stretch method targets is indeed the typical situation in tabular machine learning.

**Contrast with Image Data.** To provide context for these tabular data characteristics, we compare against four standard image datasets: MNIST (LeCun & Cortes, 2010), Fashion-MNIST (Xiao et al., 2017), CIFAR-10, and CIFAR-100 (Krizhevsky et al., 2009), which represent canonical benchmarks in computer vision. These image datasets exhibit dramatically different marginal patterns: their standard deviations cluster tightly between 0.28 and 0.39 and skewness ranges from 0.12 to 1.75. This stark contrast reveals a fundamental structural difference between tabular and image data. Images have an inherently regular structure: pixels are arranged on a fixed spatial grid where each pixel is uniformly spaced from its neighbors, and pixel intensities are bounded within fixed ranges. This structural regularity results in naturally smooth and consistent marginal slope patterns. In contrast, tabular data lacks such constraints. Numeric features can have arbitrary distributions and scales, samples can be arbitrarily close in feature space, and targets (especially for regression) can span wide ranges, leading to non-uniform marginal patterns with localized sharp variations.

**Meaningful Marginal Contributions Across Feature Compositions.** The pattern of non-uniform marginal variation appears consistently across all three feature type categories in our benchmark (balanced, category dominant, and numeric dominant). This consistency arises because the strength and non-uniformity of numeric features' marginal effects on the target are independent of how many numeric features are present or their proportion in the dataset. Even in category-dominant datasets where numeric features are the minority, we observe substantial variation (e.g., `socmob`: std=2.03, skewness=6.26; `thyroid-dis`: std=0.94, skewness=1.20). Our supervised stretch method adaptively identifies and smooths these non-uniform patterns where they exist, which explains why the method remains effective even in datasets where numeric features constitute a small proportion of the total feature set. The benefit of proper transformation is determined by the presence of non-uniform marginal patterns in the numeric features themselves, not by their count or proportion.

**Implications for Method Design.** This analysis validates the core premise of our supervised stretch framework. When features exhibit sharp marginal variations, the method identifies high-gradient regions and allocates more transformed space to them, effectively smoothing the target function where smoothing is most needed. When features have relatively flat marginal relationships in certain regions, the optimization naturally allocates minimal width variation, having negligible impact. The method automatically adapts its transformation strength to match the actual signal characteristics of each feature. The consistent improvements across our diverse benchmark reflect the method's ability to address these pervasive non-uniform marginal patterns.

In summary, this quantitative analysis demonstrates that non-uniform marginal effects with localized sharp variations are a predominant characteristic of tabular data, not an exception. This finding validates both the motivation for and the design of our supervised stretch transformation framework.

Table 6: Marginal distribution metrics (Std and Skewness)

| Dataset | Task Type | Std | Skewness |
|---------|-----------|-----|----------|
| **Balanced Datasets** | | | |
| Diamonds | Regression | 2.1275 | 5.52 |
| E-CommereShippingData | Binary | 1.3061 | 10.28 |
| Fitness_Club_c | Binary | 1.2097 | 0.60 |
| Kaggle_bike_sharing_demand_challange | Regression | 1.0102 | 2.47 |
| NHANES_age_prediction | Regression | 0.9773 | 1.57 |
| Shop_Customer_Data | Regression | 2.6387 | 6.50 |
| compass | Binary | 0.9258 | 1.02 |
| estimation_of_obesity_levels | Multiclass | 0.7353 | -0.05 |
| **Category Dominant Datasets** | | | |
| archive_r56_Portuguese | Regression | 1.1955 | 1.33 |
| okcupid_stem | Multiclass | 1.1132 | 1.44 |
| socmob | Regression | 2.0314 | 6.26 |
| thyroid-dis | Multiclass | 0.9361 | 1.20 |
| **Numeric Dominant Datasets** | | | |
| ASP-POTASSCO-classification | Multiclass | 0.4705 | -1.27 |
| Ailerons | Regression | 1.3136 | 3.48 |
| Bias_correction_r_2 | Regression | 12.5727 | 53.98 |
| Click_prediction_small | Binary | 1.0574 | 0.72 |
| Contaminant-detection-11.0GHz | Binary | 1.0746 | 0.66 |
| FOREX_audjpy-day-High | Binary | 1.0024 | 0.03 |
| FOREX_audsgd-hour-High | Binary | 0.8862 | 0.09 |
| IEEE80211aa-GATS | Regression | 13.4710 | 61.56 |
| Intersectional-Bias-Assessment | Binary | 1.1410 | 0.33 |
| Job_Profitability | Regression | 10.1227 | 73.20 |
| Mobile_Price_Classification | Multiclass | 0.8191 | 0.74 |
| VulNoneVul | Binary | 2.3204 | 3.53 |
| Waterstress | Binary | 1.0665 | 0.14 |
| electricity | Binary | 1.1502 | 1.65 |
| htru | Binary | 3.3842 | 4.15 |
| ibm-employee-performance | Binary | 1.5927 | 1.21 |
| internet_firewall | Multiclass | 20.1475 | 53.98 |
| jungle_chess_2pcs_endgame | Multiclass | 0.6012 | 0.50 |
| kc1 | Binary | 1.1874 | 0.64 |
| optdigits | Multiclass | 0.5452 | 2.06 |
| page-blocks | Multiclass | 2.0516 | 2.73 |
| pol | Binary | 1.6962 | 3.22 |
| pol_reg | Regression | 1.6065 | 2.54 |
| pole | Regression | 1.5757 | 2.30 |
| wine | Binary | 1.0016 | 1.02 |
| yeast | Multiclass | 0.8551 | 1.20 |
| **Image Datasets** | | | |
| MNIST | Multiclass | 0.353687 | 0.12 |
| Fashion-MNIST | Multiclass | 0.391164 | 0.94 |
| CIFAR-10 | Multiclass | 0.380173 | 1.47 |
| CIFAR-100 | Multiclass | 0.276569 | 1.75 |

## G   AGGREGATED PERFORMANCE WITH STATISTICAL DISPERSION

Table 7 expands upon the main results (Table 1) by providing the raw performance metrics with statistical dispersion. For each model-transformation configuration, we report the Mean ± Std. Here, the "Std" represents the average standard deviation across all datasets within the task category (Classification or Regression).

Specifically, for every single dataset-model pair, we compute the standard deviation of the test metric over 15 independent random seeds. The values reported in the table are the arithmetic means of these individual standard deviations. Note that while this table provides absolute metrics, we refer readers to Figure 2 (Sensitivity Analysis), Figure 3 (Pairwise Win-Rates) and Appendix B for comprehensive statistical comparisons that rigorously account for cross-dataset heterogeneity.

Table 7: Performance of non-trainable transformations (mean ± std).

| Metric | Model | Transformations | | | | | | | | |
|---|---|---|---|---|---|---|---|---|---|---|
| | | Stretch Sup. | Stretch Unsup. | Minmax | Ple | Ple T | Quantile | RS-SC | Standard | YJ |
| Avg Acc | ftt | **0.8274** ± 0.0144 | 0.8244 ± 0.0098 | 0.8052 ± 0.0095 | 0.8262 ± 0.0080 | 0.8261 ± 0.0093 | 0.8272 ± 0.0109 | 0.8191 ± 0.0116 | 0.8162 ± 0.0150 | 0.8212 ± 0.0107 |
| | mlp | 0.8209 ± 0.0116 | 0.8245 ± 0.0084 | 0.8152 ± 0.0070 | **0.8275** ± 0.0065 | 0.8217 ± 0.0071 | 0.8160 ± 0.0076 | 0.8235 ± 0.0088 | 0.8193 ± 0.0078 | 0.8222 ± 0.0074 |
| | mlp_plr | 0.8257 ± 0.0057 | 0.8264 ± 0.0089 | 0.8210 ± 0.0070 | 0.8212 ± 0.0080 | 0.8213 ± 0.0099 | **0.8271** ± 0.0087 | 0.8228 ± 0.0084 | 0.8212 ± 0.0083 | 0.8214 ± 0.0109 |
| | realmlp | 0.8373 ± 0.0061 | 0.8376 ± 0.0053 | **0.8387** ± 0.0062 | 0.8331 ± 0.0066 | 0.8345 ± 0.0068 | 0.8370 ± 0.0066 | 0.8383 ± 0.0062 | 0.8327 ± 0.0056 | 0.8339 ± 0.0060 |
| | resnet | **0.8257** ± 0.0078 | 0.8239 ± 0.0109 | 0.8145 ± 0.0128 | 0.8236 ± 0.0075 | 0.8232 ± 0.0075 | 0.8163 ± 0.0080 | 0.8246 ± 0.0072 | 0.8189 ± 0.0067 | 0.8201 ± 0.0086 |
| Avg $R^2$ | ftt | **0.6911** ± 0.0087 | 0.6809 ± 0.0068 | 0.6663 ± 0.0135 | 0.6799 ± 0.0221 | 0.6671 ± 0.0121 | 0.6813 ± 0.0096 | 0.6726 ± 0.0126 | 0.6815 ± 0.0052 | 0.6669 ± 0.0050 |
| | mlp | **0.6304** ± 0.0182 | 0.6060 ± 0.0260 | 0.6279 ± 0.0301 | 0.6077 ± 0.1499 | 0.5877 ± 0.0410 | 0.6190 ± 0.0208 | 0.6192 ± 0.0200 | 0.5909 ± 0.1610 | 0.5453 ± >0.20 |
| | mlp_plr | 0.6983 ± 0.0150 | 0.6963 ± 0.0155 | 0.6591 ± 0.0139 | 0.6859 ± 0.0089 | **0.7020** ± 0.0091 | 0.6689 ± 0.0199 | 0.6809 ± 0.0154 | 0.6754 ± 0.0132 | 0.6489 ± 0.0178 |
| | realmlp | **0.7362** ± 0.0137 | 0.6846 ± 0.0147 | 0.6791 ± 0.0100 | 0.7357 ± 0.0099 | 0.7234 ± 0.0129 | 0.7066 ± 0.0179 | 0.6888 ± 0.0166 | 0.6858 ± 0.0092 | 0.6832 ± 0.0304 |
| | resnet | 0.6507 ± 0.0120 | 0.6312 ± 0.0248 | **0.6547** ± 0.0094 | 0.6386 ± >0.20 | 0.6307 ± 0.0182 | 0.6254 ± 0.0417 | 0.6399 ± 0.0085 | 0.6409 ± 0.0098 | 0.5421 ± >0.20 |

## H   NOISE ROBUSTNESS ANALYSIS

To evaluate the applicability of our method in noisy environments, we perform a systematic robustness analysis covering both classification and regression tasks.

### H.1   EXPERIMENT SETUP

We include four diverse datasets from our benchmark: `Contaminant-detection`, `yeast`, `ibm-employee-performance` (Classification), and `NHANES-age-prediction` (Regression). We evaluate the performance of Supervised Stretch and Unsupervised Stretch against Standardization, PLE, and PLE-T across five neural architectures.

**Noise Injection Protocol**: We tailor the noise injection to the task type:

- **Classification**: We apply Symmetric Label Flipping. For a noise level $\eta$, we randomly flip the labels of $\eta \times 100\%$ of the training samples.
- **Regression**: We add Gaussian noise $\epsilon \sim \mathcal{N}(0, \sigma_{noise}^2)$ to the targets, where $\sigma_{noise} = \eta \cdot \sigma_{target}$ (i.e., noise is scaled relative to the target's standard deviation).

Validation and test sets remain clean to evaluate signal recovery.

**Evaluation Metric (Mean Rank)**: To enable a unified comparison across datasets with different metrics (Accuracy vs $R^2$) and baselines, we use **Mean Rank**. For each noise level, we rank the 5 transformations for every (dataset, model) pair ($1 = $ Best, $5 = $ Worst). We report the average rank aggregated across all 20 combinations (4 datasets $\times$ 5 models). Error bars represent the standard deviation of ranks, indicating consistency across architectures.

### H.2   RESULTS AND DISCUSSION

Figure 10 presents the Overall Mean Rank aggregated across all datasets and models, while Figures 11a-12b provide detailed rank trajectories for individual datasets. The results reveal a compelling trend regarding the robustness of Supervised Stretch:

- **Supervised Stretch Demonstrates Resilience**: Contrary to concerns that leveraging target information might lead to overfitting noise, Supervised Stretch (Red) maintains a competitive performance profile throughout the noise spectrum. As shown in Figure 10, its mean rank fluctuates within a strong range and does not exhibit the systematic degradation seen in some baselines. Notably, even at the most extreme noise level (50%), it achieves the best mean rank among all methods.

- **Comparative Stability of Baselines**: Standardization (Blue) remains a stable baseline. Unsupervised Stretch (Purple) maintains a consistent profile; notably, at extreme noise levels ($\eta = 0.5$), it secures the 3rd rank, trailing PLE only marginally. This is a noteworthy result given that Unsupervised Stretch achieves this competitive robustness without the dimensional expansion ($O(1)$ vs $O(T)$) required by PLE, making it a highly efficient fallback when target reliability is low.

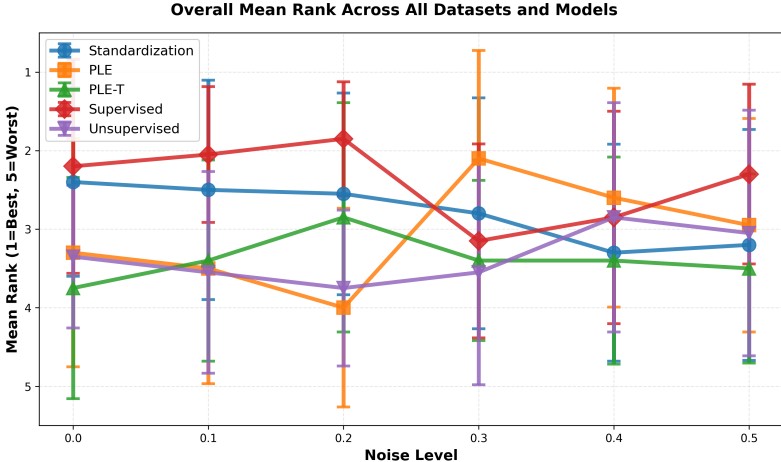

Figure 10: **Overall Mean Rank Analysis under Label Noise.** The rank is computed for each (dataset, model) pair and averaged across **20 combinations** (4 **datasets** ×5 **models**). Lower rank indicates better performance. **Supervised Stretch (Red)** demonstrates superior robustness, improving its relative ranking to become the top-performing method as noise intensity increases to 50%. Error bars denote the standard deviation of ranks across models and datasets.

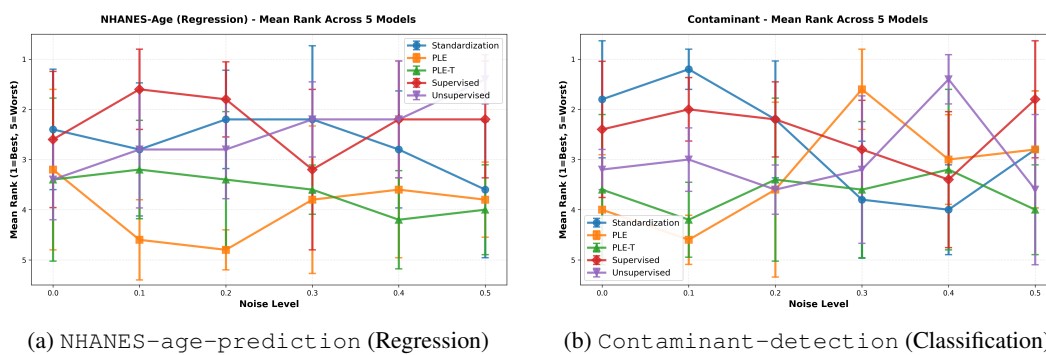

(a) NHANES-age-prediction (Regression)   (b) Contaminant-detection (Classification)

Figure 11: **Detailed Mean Rank Trajectories (1/2).** Mean rank across 5 models for NHANES-age and Contaminant datasets under varying noise levels.

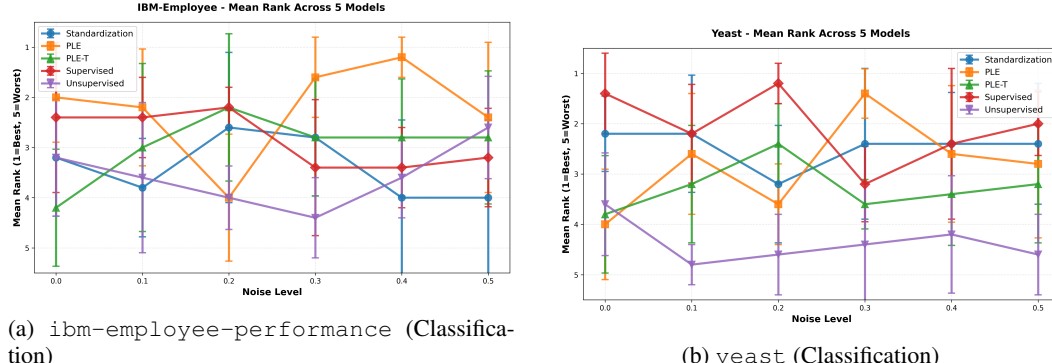

(a) `ibm-employee-performance` (Classification)

(b) `yeast` (Classification)

Figure 12: **Detailed Mean Rank Trajectories (2/2).** Mean rank across 5 models for IBM-Employee and Yeast datasets under varying noise levels.

## I    STATEMENT ON THE USE OF LARGE LANGUAGE MODELS

In adherence to the ICLR 2026 policy, we hereby state that the scientific contributions and the core text of this paper were conceived and written entirely by the human authors.

We utilized a large language model (LLM) as a writing assistance tool for the purpose of improving the manuscript's clarity and readability. The use of the LLM was limited to tasks such as:

- Correcting grammatical errors and refining punctuation.
- Improving sentence fluency and structure.
- Rephrasing sentences to enhance clarity, while preserving the original meaning.
- Ensuring consistent terminology throughout the document.

The authors reviewed, edited, and approved all suggestions made by the LLM. The final content, including all scientific claims and the phrasing of the text, remains the full responsibility of the human authors.

