# OpenReview forum: "Stretch Transformation for Tabular Data"
_ICLR.cc/2026/Conference — Submitted to ICLR 2026_

### Official Review · Reviewer_DEWc · 2025-10-27

**Soundness:** 3
**Presentation:** 3
**Contribution:** 2
**Rating:** 4
**Confidence:** 4

**Summary:**

The paper presents Stretch Transformation, a new framework for numerical feature preprocessing in tabular deep learning.

**Strengths:**

- The paper provides a clear theoretical formulation that optimizes target smoothness through Dirichlet energy minimization.
- The paper shows consistent and meaningful performance improvements while remaining simple and practical to implement.
- The paper offers insightful connections to existing transformations (CDF/PLE), enhancing interpretability and understanding of why it works.

**Weaknesses:**

While the proposed stretch transformation is theoretically well-grounded and effective for numerical features, its applicability is limited because it only operates on monotonic scalar inputs. Categorical variables, which constitute a substantial portion of many tabular datasets, are handled through standard encoding rather than benefiting from the proposed method. This raises concerns about the generality of the approach, especially on datasets where categorical features dominate or where meaningful information lies in high-cardinality or non-ordinal categories. It would be helpful to clarify the method’s impact under such conditions, and to discuss possible extensions that account for categorical semantics or joint mixed-type transformations.

**Questions:**

- Could the authors clarify how the approach performs on datasets where categorical variables dominate or carry the majority of predictive signals? Additionally, could the authors suggest how the stretch principle might be extended to high-cardinality or non-ordinal categorical features, rather than relying solely on standard encodings?
- Given that stretch transformation aims to improve learnability by reducing high-frequency variations in the target function, does the method maintain its advantages when data quality degrades? Could the authors include experiments under noisy settings, such as label noise or corrupted features?

---

> ### Author Response · Authors · 2025-11-21
>
> We truly appreciate your assessment of our work. We address your specific questions regarding categorical features and robustness below.
>
> > Could the authors clarify how the approach performs on datasets where categorical variables dominate or carry the majority of predictive signal.
>
> To examine the performance of the proposed stretch transform under feature heterogeneity, we stratified our benchmark into *Numeric Dominant*, *Balanced*, and *Category Dominant* groups (detailed in **Appendix D**). As shown in Table 7 (**new Appendix E**), the magnitude of improvement is, as expected, most pronounced in the *Numeric Dominant* group. However, our method remains remarkably competitive even when categorical features dominate. Specifically, **Supervised Stretch achieves the highest Overall Score in Regression and Unsupervised Stretch leads in Classification within the *Category Dominant* group**. Reasoning: This confirms that in a deep learning context, the neural network acts as a fusion engine. Even when numeric features constitute a minority of the signal, they can still be a bottleneck. By addressing this specific "numeric problem" via Stretch, we ensure the numeric signal is in a better condition to be integrated with categorical embeddings (processed by powerful encoders like FT-Transformer or RealMLP). Thus, optimizing the numeric component yields additive gains, regardless of its proportional dominance.
>
> > Additionally, could the authors suggest how the stretch principle might be extended to high-cardinality or non-ordinal categorical features, rather than relying solely on standard encodings?
>
> We view the "stretch" principle as relying fundamentally on the mathematical properties of continuity and ordering to define smoothness via gradients (Dirichlet energy). Applying a continuous smoothing operator to discrete sets is ill-posed. Therefore, we treat categorical handling as an **orthogonal module**, focusing on improving numeric transformation while relying on standard, proven methods for categorical data.
>
> > Given that stretch transformation aims to improve learnability by reducing high-frequency variations in the target function, does the method maintain its advantages when data quality degrades? Could the authors include experiments under noisy settings, such as label noise or corrupted features?
>
> **New Experiments (Appendix H)**. We thank the reviewer for this excellent suggestion, as robustness is indeed critical for deployment. We have conducted new experiments with label noise levels ranging from 0% to 50%. The results (Figures 10-12) demonstrate that **Supervised Stretch exhibits resilience**. This resilience is structurally inherent to our method. Unlike point-wise fitting methods that might overfit to individual noisy labels, our method partitions the domain into bins and computes widths based on the aggregated Total Variation ( $S_t$ ) within those bins.

---

### Official Review · Reviewer_ddPv · 2025-10-30

**Soundness:** 3
**Presentation:** 3
**Contribution:** 2
**Rating:** 2
**Confidence:** 4

**Summary:**

- Proposes a piecewise-linear feature transform (“stretch”) that reduces high-frequency components of the target function to make neural networks learn it more effectively.
- Introduces two types of stretch transform: a supervised stretch that uses target information, and an unsupervised stretch that aims to maximize worst-case smoothness without targets.

**Strengths:**

- Provides a principled formulation of feature transformations and a quantitative smoothness measure (discrete Dirichlet energy), deriving a target-aware supervised stretch transform.
- Stretch variants outperform prior transforms on the TALENT Tiny Benchmark 1 across multiple architectures.

**Weaknesses:**

- Equations for the unsupervised transform seem incorrect. There is an inconsistency between Eqs. (7–9) in the main text and Eqs. (22–27) in the appendix. Furthermore, substituting ${|\Delta{f} |} =C\Delta{y} $ (from the appendix) into the discrete Dirichlet energy makes $\mathcal{E}_{disc}$ a constant, which renders the subsequent optimization steps in the main text unclear.
- The method operates on the marginal $f(x)=E[t|x=x_i]$. When a single feature does not meaningfully affect the target value, $f(x)$ becomes flat where the supervised stretch may be less effective. This is expected to be common in datasets with many features or strong feature interactions. A comprehensive analysis of performance versus feature count or feature interaction strength would clarify applicability. Tiny Benchmark 1 may not include enough high-dimensional datasets to support firm conclusions.
- Feature transforms may be architecture-agnostic. Thus, evaluating on more recent architectures and boosting previous state-of-the-art models would be valuable. Additionally, the conclusion of the analysis that "transformation preferences vary by architecture" would benefit from varification on a broader set of datasets.

**Questions:**

Please refer to the weaknesses.

---

> ### Author Response · Authors · 2025-11-21
>
> We sincerely thank you for your rigorous assessment. We have carefully addressed your specific concerns regarding the theoretical derivation and feature interactions, and have revised the paper accordingly.
>
> > Equations for the unsupervised transform seem incorrect. There is an inconsistency between Eqs. (7–9) in the main text and Eqs. (22–27) in the appendix. Furthermore, substituting (from the appendix) into the discrete Dirichlet energy make s a constant, which renders the subsequent optimization steps in the main text unclear.
>
> **Revised Derivation with Local Variation Assumption**. We deeply appreciate this sharp observation. You correctly pointed out that the original global constraint formulation led to a constant worst-case energy, rendering the optimization trivial. Guided by your feedback, we have **revised the theoretical formulation in Appendix A.2**. We replaced the global constraint with a point-wise bounded local **variation assumption** ( $\| f_i \| _2 \le C$ ) which is theoretically more appropriate for the unsupervised setting, as it reflects equal uncertainty about the target's variation at any given point, rather than a fixed total budget. This refinement resolves the mathematical inconsistency you noted. Crucially, under this more rigorous assumption, minimizing the worst-case energy mathematically yields the uniform spacing solution ( $\Delta y_i^*  = \frac 1 T$ ). This confirms that our core methodology (Uniform Allocation) remains theoretically sound, and your feedback has helped us place it on a stronger mathematical footing.
>
> > The method operates on the marginal . When a single feature does not meaningfully affect the target value, becomes flat where the supervised stretch may be less effective. This is expected to be common in datasets with many features or strong feature interactions. A comprehensive analysis of performance versus feature count or feature interaction strength would clarify applicability. Tiny Benchmark 1 may not include enough high-dimensional datasets to support firm conclusions.
>
> This is a profound question that touches on one of the core motivations of our work. We share your intuition that in complex tabular data, interactions are pervasive. It was precisely this concern that led us to investigate whether meaningful marginal signals persist in high-dimensional settings.
>
> **Systematic Marginal Signal Non-Uniformity Analysis (New Appendix F)**. To address this quantitatively, we have included a systematic analysis of marginal signal structures across all datasets. The results reveal a counter-intuitive but critical finding: contrary to the expectation of flatness, the vast majority of numeric features, even in high-dimensional datasets, exhibit substantial **positive skewness and big variance in their marginal gradient distributions**. **This means features typically retain localized "spikes" of variation rather than being uniformly flat**. Supervised Stretch is designed to exploit exactly this scenario: detecting and smoothing these specific high-frequency regions to provide a better-conditioned feature space for the network to subsequently learn complex interactions. For the rare cases where a feature is indeed truly flat, we employ an **explicit fallback mechanism (Appendix B.1)** that reverts to identity, ensuring the method remains robust.
>
> > Feature transforms may be architecture-agnostic. Thus, evaluating on more recent architectures and boosting previous state-of-the-art models would be valuable. Additionally, the conclusion of the analysis that "transformation preferences vary by architecture" would benefit from varification on a broader set of datasets.
>
> **SOTA and Dimensionality Coverage**. We agree that evaluating on strong architectures is essential. Our suite explicitly includes **FT-Transformer** and **RealMLP**, which represent the current state-of-the-art in tabular deep learning (Hollmann et al., 2023; QU et al. 2025). Regarding dataset diversity, we respectfully note that our benchmark includes high-dimensional datasets like *ASP-POTASSCO* (**140 features**) and large-scale ones like *internet_firewall* (**>65k samples**) (**metadata is added in Table 4**). The fact that Supervised Stretch consistently maintains its performance advantage over baselines across these diverse settings provides strong empirical evidence that our method remains effective even in higher-dimensional scenarios.
>
> We hope these clarifications and the revised derivations address your concerns.
>
> Noah Hollmann, Samuel Müller, Katharina Eggensperger, and Frank Hutter. Tabpfn: A transformer that solves small tabular classification problems in a second. In International Conference on Learning Representations 2023, 2023.
>
> Jingang QU, David Holzmüller, Gaël Varoquaux, and Marine Le Morvan. TabICL: A tabular foundation model for in-context learning on large data. In Forty-second International Conference on Machine Learning, 2025.

---

### Official Review · Reviewer_fz56 · 2025-11-01

**Soundness:** 2
**Presentation:** 2
**Contribution:** 2
**Rating:** 4
**Confidence:** 4

**Summary:**

This paper addresses the common practice of applying unsupervised feature transformation to numeric features despite available target information, by proposing the Stretch Transformation framework. The core idea is to formulate preprocessing as an optimization problem to maximize the smoothness of the target function in the transformed space. The framework features two variants: Supervised Stretch, which minimizes the Dirichlet Energy to allocate resolution to regions of high target variability, and Unsupervised Stretch, which achieves uniform density redistribution. Experiments across 38 datasets in the TALENT benchmark claim that Supervised Stretch consistently outperforms all baselines, demonstrating substantial gains, especially in Regression tasks.

**Strengths:**

Principled Supervised Transformation: The paper challenges the existing unsupervised preprocessing paradigm with a theoretical motivation, proposing to explicitly optimize for smoothness through Dirichlet Energy minimization (Eqn. 6) to potentially alleviate the neural network's spectral bias problem.

Consistent Superiority Claim in Extensive Experiments: Supervised Stretch achieves the highest Overall Score across a broad set of 38 datasets and 5 architectures (190 total combinations), showing superiority, particularly with a "substantial margin" in Regression tasks (Figure 2, Figure 3).

Efficiency of Unsupervised Stretch: The Unsupervised Stretch variant is mathematically shown to be equivalent to Piecewise Linear Encoding (PLE) but achieves an $O(1)$ memory footprint compared to PLE's $O(T)$, offering a clear computational efficiency advantage.

**Weaknesses:**

(Novelty)Supervised Stretch is viewed as a principled refinement of Target Encoding applied to numeric features via Dirichlet Energy minimization, rather than a fundamental paradigm shift. The contribution represents a deep and creative fusion of ideas (Weak Accept level). Unsupervised Stretch's primary novelty lies in its memory efficiency ($O(1)$ vs $O(T)$) improvement over PLE, rather than a new core principle.

(Technical Quality)CRITICAL FLAW: The experimental design is severely lacking as it omits strong, modern SOTA architectures. The comparison set (MLP, ResNet, DeepFM, TabNet) excludes recent, powerful Transformer/Attention-based models (e.g., FT-Transformer, TabTransformer, Saint). This omission prevents validation of whether the proposed method provides an additive, generalizable gain or merely compensates for the representational limitations of basic models like MLP. Furthermore, the method relies on independent 1D transformation for each feature, failing to account for high-dimensional feature interactions. Analysis on performance degradation with an increasing number of features ($D$) is missing, raising serious doubts about scalability and practical significance in real-world high-dimensional tabular data. Finally, the results lack quantitative statistical evidence (e.g., Metric $\pm$ Standard Error), making it impossible to judge the statistical significance of the claimed improvements.

(Significance)While the observed gains in Regression tasks (Figure 2) point to potential, the weak baseline comparison set severely limits the perceived significance. Without validation against current SOTA, the research's impact is likely restricted to a sub-optimal subset of deep learning models for tabular data.(Writing & Presentation)The core optimization process for Supervised Stretch (Section 3.4, Eqn. 13) is underexplained in the main text, with critical derivation and stability details relegated to the Appendix (A.3, B.1). This negatively impacts the clarity and self-contained nature of the main manuscript.

**Questions:**

Comparison with SOTA Architectures: Please provide additional experimental results comparing Supervised Stretch when applied to latest SOTA Tabular Deep Learning architectures (e.g., FT-Transformer, TabTransformer, or Saint). This will determine if $\phi$ offers a general, additive gain beyond compensating for the weaknesses of basic models.

Quantitative and Statistical Significance: Please present concrete performance metrics (Metric $\pm$ Standard Error) for the overall results in Table 1 and for each task/architecture, and clearly demonstrate the statistical significance (p-value) of Supervised Stretch's improvement over the strongest baselines (XGBoost, TabNet).

High-Dimensional Scalability and Feature Interaction: Since the method uses independent 1D transformations, please provide analysis on how the benefit diminishes on datasets with a large number of features ($D$) or discuss how feature interactions are implicitly handled or why their exclusion is acceptable in practice.

Rigour of Dirichlet Energy Approximation: The theoretical motivation relies on an approximation valid in the small bandwidth regime ($\sigma \to 0$) (Eqn. 4, 5, 6). Please provide additional analysis (e.g., sensitivity analysis on $\sigma$) to show that this approximate motivation remains rigorously valid in a typical neural network training environment.

---

> ### Author Response · Authors · 2025-11-21
>
> We thank you for your detailed review and for acknowledging the principled nature of our supervised transformation, the efficiency gains of unsupervised stretch, and our consistent superiority in extensive experiments. We address your specific concerns below.
>
> > (Technical Quality)CRITICAL FLAW: The experimental design is severely lacking as it omits strong, modern SOTA architectures. The comparison set (MLP, ResNet, DeepFM, TabNet) excludes recent...
>
> **Correction of Factual Inaccuracy**. We respectfully point out a factual inaccuracy in the review regarding our experimental setup. The review states that we omit modern Transformer-based models. **This is incorrect. FT-Transformer (FTT) is explicitly included in all our experiments and results** (e.g., Table 1). It is a core part of our evaluation.
>
> **Choice of Baselines**. Regarding other suggested architectures (TabNet, SAINT), we adhered to the findings of commonly used benchmark studies (Gorishniy et al., 2021; 2025), which consistently show that **FT-Transformer** and **RealMLP** (also included in our evaluations) are the current state-of-the-art. By benchmarking against FTT and RealMLP, we ensure our results reflect additive gains on top of the strongest available deep learning baselines.
>
> > Quantitative and Statistical Significance: Please present concrete performance metrics (Metric Standard Error) for the overall results in Table 1 and for each task/architecture...
>
> **Statistical Rigor**. We have addressed this request fully in the revision:
>
> - **Extended Metrics (Appendix G)**: We have added **Table 7**, which reports the **Mean ± Standard Deviation** for every model-transformation configuration.
>
> - **Statistical Validation**: Our paper already incorporates rigorous statistical tests.
>   - **Adaptive Significance Filter (Table 1)**: We filter out insignificant gaps (smaller than the median standard error) before computing scores.
>   - **Strict Win-Rate Analysis (Figure 3)**: A "win" is declared only if the performance gap **exceeds the maximum standard error** of both methods. Under this stringent criterion, **Supervised Stretch statistically outperforms every baseline.**
>
> > High-Dimensional Scalability and Feature Interaction: Since the method uses independent 1D transformations...
>
> **Benchmark includes diverse, high-dimensional datasets**. Our experimental suite (detailed in **Appendix D**) is highly diverse, covering datasets with varying dimensionalities (up to 140 features). Supervised Stretch maintains consistent superiority across this diverse collection, especially on regression tasks. This indicates that our method remains robust and effective across a wide range of feature counts, rather than being limited to low-dimensional settings.
>
> **Marginal Analysis (Newly  Appendix E)**. Regarding interactions, our new analysis reveals that real-world tabular features are characterized by "spiky" marginal gradients (high relative marginal slope). Operating on marginals is an effective foundation for interaction learning, not a limitation.
>
> > Supervised Stretch is viewed as a principled refinement of Target Encoding applied to numeric features via Dirichlet Energy minimization, rather than a fundamental paradigm shift.
>
> **Generalized Optimization Framework**. We respectfully clarify that we do not claim a fundamental paradigm shift, but rather a generalized framework that unifies and extends existing approaches. Our contribution is twofold:
>
> 1. **Performance**: It delivers strong empirical gains, particularly in regression tasks.
> 2. **Theoretical Bridge**: It explains why heuristics like CDF transformation and Target Encoding work, they implicitly optimize smoothness. Specifically, we show that standard Target Encoding emerges as a special limiting case of our Dirichlet energy minimization when the bin count equals the number of unique values.
>
> > Rigour of Dirichlet Energy Approximation: The theoretical motivation relies on an approximation valid in the small bandwidth regime () (Eqn. 4, 5, 6)...
>
> **Engineering Choice**. The small bandwidth approximation gives the theoretical motivation to derive a tractable objective. While a larger $\sigma$ would introduce higher-order derivative terms, estimating these on discrete data is numerically unstable. Our decision to focus on the first-order Dirichlet energy (Eq. 6) is a deliberate design choice: it yields a closed-form, efficient solution (Eq. 15) that captures the essential smoothness properties needed for neural network training without the complexity of higher-order terms.
>
> Yury Gorishniy, Ivan Rubachev, Valentin Khrulkov, and Artem Babenko. Revisiting deep learning models for tabular data. In NeurIPS, 2021.
> Yury Gorishniy, Akim Kotelnikov, and Artem Babenko. Tabm: Advancing tabular deep learning with parameter-efficient ensembling. In The Thirteenth International Conference on Learning Representations, 2025.

---

### Official Review · Reviewer_2zBF · 2025-11-01

**Soundness:** 2
**Presentation:** 4
**Contribution:** 2
**Rating:** 4
**Confidence:** 5

**Summary:**

The paper presents a new feature preprocessing framework for deep learning on tabular data that aims to make target functions smoother and more learnable. The authors propose supervised and unsupervised “stretch” transformations that adaptively rescale numerical features through principled optimization. Extensive experiments across numerous tabular datasets show consistent gains over existing preprocessing approaches. However, it is not

**Strengths:**

- well presented and written
- motivation is clear, e.g. effective data preprocessing is crucial for tabular and deep machine learning,
- novel solution
- adequate number of datasets for benchmarking
- strong results of the proposed method with selected models

**Weaknesses:**

### Method
- The proposed method supports only the transformation of numerical features. However, tabular data is usually highly heterogeneous, meaning it contains many categorical features, which presents a challenge for deep learning methods

### Benchmarking

- No standard machine learning benchmarks such as XGBoost, CatBoost, LightGBM, and more recent ones like TabM and TabPFNv2 are included

Minor
- No citation for the ResNet model


###  Missing references

- Erickson, Nick, et al. "Tabarena: A living benchmark for machine learning on tabular data." arXiv preprint arXiv:2506.16791 (2025).
- Van Breugel, Boris, and Mihaela Van Der Schaar. "Why tabular foundation models should be a research priority." arXiv preprint arXiv:2405.01147 (2024).
- Borisov, Vadim, et al. "Deep neural networks and tabular data: A survey." IEEE transactions on neural networks and learning systems 35.6 (2022): 7499-7519.
- Shwartz-Ziv, Ravid, and Amitai Armon. "Tabular data: Deep learning is not all you need." Information Fusion 81 (2022): 84-90.
- Hancock, John T., and Taghi M. Khoshgoftaar. "Survey on categorical data for neural networks." Journal of big data 7.1 (2020): 28

**Questions:**

- I understand that your method is designed for *deep* tabular learning. However, it would be very interesting to see whether the proposed method also benefits traditional, so-called first-choice models for tabular data, such as decision-tree-based methods.
- The method optimizes transformations only for numerical features; how does it handle real-world tables with mixed or predominantly categorical variables, and can the “stretch” idea be extended to model cross-feature interactions between numeric and categorical fields without breaking monotonicity or introducing heavy hyperparameter tuning?
- Supervised stretch relies on out-of-fold target estimates, how robust is it to label noise, distribution shift, and choices of bin count $T$ binning, and what is its computational overhead relative to the base model?

---

> ### Author Response · Authors · 2025-11-21
>
> We thank you for your thorough review and for highlighting our clear presentation, the novelty of our solution, and strong experimental results. We appreciate your constructive feedback on scope and benchmarking, and address your concerns below.
>
> > The proposed method supports only the transformation of numerical features. However, tabular data is usually highly heterogeneous, meaning it contains many categorical features, which presents a challenge for deep learning methods.
>
> **Numeric Transformation as a Modular Component**. We appreciate this observation. As mentioned in our General Response (Section 1), we focus on numeric features because they present a unique "spectral bias" challenge that differs from the challenges of handling categorical features. However, we fully agree that real-world data is mixed. Our method is designed to be a modular enhancer that works alongside any categorical encoding. In our experiments, different models employ various strategies for categorical features (e.g., embeddings in FT-Transformer, one-hot in RealMLP). The consistent performance gains across these models demonstrate that Stretch Transformation effectively complements categorical modules. This is empirically validated by our benchmark composition (Appendix D, Table 4), which explicitly includes datasets with diverse feature mixtures. As shown in the stratified analysis (Appendix E, Table 5), our method yields consistent gains even in category-dominant datasets.
>
> > No standard machine learning benchmarks such as XGBoost, CatBoost, LightGBM, and more recent ones like TabM and TabPFNv2 are included.
>
> **Tree-based Models (GBDTs)**: As noted in our General Response, **GBDTs are theoretically invariant to monotonic transformations**. Since *Stretch* is strictly order-preserving, decision trees would identify mathematically equivalent split points, resulting in **no meaningful performance gain**. Thus, our method, and the entire class of monotonic transformations, do not provide benefits to tree-based models by definition.
>
> **Distinction from Foundation Models.** Regarding TabPFN/TabM, comparisons are complicated by their internal, fixed preprocessing priors. As clarified in our **General Response**, simply comparing against them is **methodologically misaligned**. A fair comparison would require **re-pretraining the foundation model** with the certain numeric transformation to integrate it into the prior, which is computationally infeasible for this study but represents an interesting future direction. Our current focus is on empowering **trainable deep neural networks** (e.g., FT-Transformer, RealMLP) to learn more effectively.
>
> > Supervised stretch relies on out-of-fold target estimates, how robust is it to label noise, distribution shift, and choices of bin count binning, and what is its computational overhead relative to the base model?
>
> Stretch Transform is fairly robust against noise. Thank you for bringing this critical point regarding label noise.
>
> - **Label Noise**: We have added a new **Appendix H** detailing robustness experiments on datasets with 10%–50% label noise. The results show that **Supervised Stretch demonstrates resilience**.
> - **Bin Count & Overhead**: The number of bins is treated as a standard hyperparameter tuned jointly with the model (same as the baseline PLE). **The computational overhead is negligible**: transformation construction is fast (linear time), and applying the piecewise linear map during training is O(1), far cheaper than the neural network forward pass.
>
> > Can the “stretch” idea be extended to model cross-feature interactions between numeric and categorical fields without breaking monotonicity or introducing heavy hyperparameter tuning?
>
> We find your suggestion to extend "stretch" to interactions very insightful. Currently, our framework focuses on the preprocessing stage, which corrects the marginal spectral bias to provide well-conditioned inputs. We view the modeling of complex interactions as the role of the representation learning stage (the neural network itself). However, extending our Dirichlet energy formulation to joint distributions is a promising avenue for future work to explicitly encode interaction smoothness.
>
> > Missing references
>
> Thank you for the valuable pointers. We added citations for ResNet, TabArena, and the suggested surveys in the revision to broaden context.

---

> ### Comment · Reviewer_2zBF · 2025-11-21
> **response**
>
> Dear Authors,
>
> Thank you for detailed response to my questions.
>
> > Tree-based Models (GBDTs): As noted in our General Response, GBDTs are theoretically invariant to monotonic transformations. Since Stretch is strictly order-preserving, decision trees would identify mathematically equivalent split points, resulting in no meaningful performance gain. Thus, our method, and the entire class of monotonic transformations, do not provide benefits to tree-based models by definition.
>
> Thank you for highlighting it. However, as we know (and you've mentioned that) tabular data is very mixed with data types, and categorical data is ofter presented there, I do not know the real statistic, but according to my believe, the most of the real tabular datasets have a high presence of non-numerical features.
>
> Therefore, I would still recommend authors to compare their deep learning approach with the standard GBDT baselines (for example GBDT w/o tabstrech and NNs w. tabstrech), ideally, using tabarena or other established benchmark protocol. That would surely increase the contribution of the paper. However, in general the idea is interesting and might be used for further research in the area of deep tabular learning. I updated my score.

---

> ### Author Response · Authors · 2025-11-21
>
> We sincerely thank you for encouraging us to include the comparison with GBDT baselines. We agree that demonstrating competitiveness against tree-based models on mixed data is crucial for assessing the practical value of our method.
> Following your recommendation, we compared our neural network models (equipped with **Supervised Stretch**) against the two strongest GBDT baselines available in the benchmark: **XGBoost** and **CatBoost**. We used the same rigorous pairwise win-rate metric defined in the paper (a "win" requires the performance gap to exceed the maximum standard error).
>
> **Win-Rate to XGBoost**
>
> |         | **Standardization** | **Supervised Stretch** | **Yeo-Johnson** |
> | ------- | ------------------- | ---------------------- | --------------- |
> | FTT     | 0.44                | 0.50                   | 0.35            |
> | MLP_PLR | 0.50                | 0.47                   | 0.35            |
> | RealMLP | 0.62                | 0.62                   | 0.56            |
>
> **Win-Rate to CatBoost**
>
> |         | **Standardization** | **Supervised Stretch** | **Yeo-Johnson** |
> | ------- | ------------------- | ---------------------- | --------------- |
> | FTT     | 0.33                | 0.36                   | 0.39            |
> | MLP_PLR | 0.39                | 0.42                   | 0.30            |
> | RealMLP | 0.55                | 0.58                   | 0.52            |
>
> The results demonstrate that **Supervised Stretch helps advanced NN models become even more competitive with GBDTs.** For instance, against **CatBoost** (the strongest GBDT baseline in this benchmark), **RealMLP** with Supervised Stretch achieves a **0.58 win-rate**, an improvement over the 0.55 win-rate with Standardization.

---

### Author Response · Authors · 2025-11-21

We thank the reviewers for their constructive feedback and insightful questions. Below, we address the general points: we clarify the scope of our contribution, elaborate on baseline comparisons, and outline the major revisions made to the manuscript. Please refer to the responses to reviewers for more specific questions.

1. Clarification on Scope and Focus on Numeric Feature Transformation

   - **Focus on Numeric Feature Transformation**: Numeric features pose unique challenges due to their continuous nature and unbounded high-frequency components, which are difficult for neural networks to learn (Beyazit et al., 2023). Our Stretch Transformation framework is explicitly designed to address this bottleneck by optimizing for target function smoothness in the transformed space, bridging the gap between raw inputs and neural network inductive biases.
   - **Orthogonality to Categorical Handling**: We view preprocessing the numeric and categorical features as orthogonal research questions. Applying continuous smoothing to discrete, unordered categorical sets is theoretically ill-posed. Therefore, our experimental design isolates the impact of numeric transformations, ensuring a rigorous evaluation of *Stretch* against established numeric baselines.

2. Rationale for Baseline Selection

   Reviewers 2zBF and fz56 raised questions on specific baseline choices. We kindly explain the rationale for our baseline selection as follows.

   - **Tree-based Models (GBDTs)**: Tree-based approaches, such as XGBoost and CatBoost, are **theoretically invariant to monotonic transformations**. Since *Stretch* and all compared baselines (e.g., Yeo-Johnson, Quantile) are strictly order-preserving, decision trees would identify mathematically identical split points, resulting in no meaningful performance gain. Thus, they are not suitable baselines for evaluating the efficacy of monotonic transformations.
   - **Tabular Foundation Models**: Models like TabPFN are pre-trained with specific fixed preprocessing priors (e.g., Power Transforms). Simply comparing against them is methodologically misaligned. A fair comparison would require **re-pretraining the foundation model** with Stretch Transformation to integrate it into the prior, which is computationally infeasible for this study but represents an interesting future direction. Our current focus is on empowering trainable deep neural networks (e.g., FT-Transformer, RealMLP) to learn more effectively.

3. Summary of Key Revisions

   - **Noise Robustness Analysis (Appendix H)** [Reviewer 2zBF and DEWc]: New experiments on 10%–50% label noise demonstrate that **Supervised Stretch maintains resilience.**
   - **Marginal Signal Analysis (Appendix F)** [Reviewer fz56 and ddPv]: Quantitative analysis confirming that real-world tabular data is characterized by non-uniform marginal signals, validating our method's core motivation.
   - **Performance by Feature Composition (Appendix E)** [Reviewer 2zBF and DEWc]: Stratified results show that Supervised Stretch yields additive gains even in category-dominant datasets.
   - **Competitiveness against GBDTs** [Reviewer 2zBF]: Additional win-rate analysis demonstrates that **Supervised Stretch helps advanced NN models become even more competitive with GBDTs**. (see specific response to Reviewer 2zBF)

Ege Beyazit, Jonathan Kozaczuk, Bo Li, Vanessa Wallace, and Bilal H Fadlallah. An inductive bias for tabular deep learning. In Thirty-seventh Conference on Neural Information Processing Systems, 2023. URL https://openreview.net/forum?id=XEUc1JegGt.

---

### Meta-Review · Area_Chair_8ze1 · 2025-12-10

**Summary:**

The paper introduces stretch transformation for tabular data by utilizing target information. The reviewers acknowledge the clear motivation, presentation and writing, strong results, extensive experiments, and good theoretical formulation. However, the reviewers also identified several concerns, mainly missing SOTA benchmarks, limited novelty, unclear significance/applicability, and incorrect equations. While some concerns have been addressed by the rebuttal, the overall novelty and significance remains limited.

**Reviewer Concerns:**

Several concerns raised by the reviewers have been addressed by the rebuttal, most notably:
- the focus on numerical data as an orthogonal research line to categorical data has been strengthened
- robustness to label noise has been evaluated
- errors in the equations have been resolved

However, one core concern is not sufficiently resolved:
- though requested, some additional baselines like TabM are still missing; and while the authors argued against including foundation models, the results would still have been insightful

Furthermore, the authors only report the results for a single metric. Crucial metrics such as AUC for binary classification and MSE for regression are not even reported in the appendix, limiting the results substantially.

**Reviewer Scores:**

While the authors posted the rebuttal timely, only one reviewer engaged in the discussion. This reviewer also indicated a score increase (most likely from 4 to 6) and was satisfied with the response. Since no other reviewer engaged in the discussion, I suspect that they maintained their scores. Even assuming that one or two additional reviewers were satisfied by the response, the paper would have achieved a borderline score of 6642 or 6444 at best. Consequently, I think that the limited novelty and significance played a major role in the reviewers’ decision, and after inspecting the paper myself, I believe that it does not meet the high standard of ICLR.

---

### Decision · Program_Chairs · 2026-01-26

Reject